Synoptic revision of the fern genus Elaphoglossum Schott ex J.Sm. (Dryopteridaceae) in Madagascar, with the description of 23 new taxa, all but one endemic

http://orcid.org/0000-0002-0751-5352 Rouhan Germinal rouhan@mnhn.fr
Institut de Systématique, Évolution, Biodiversité (ISYEB), Muséum national d’Histoire naturelle, Sorbonne Université, Ecole Pratique des Hautes Etudes, Université des Antilles, CNRS ; Paris , France
Sosa Victoria
Electronic publication date: 2020 Dec 21
Publication date: 2020
Volume: 8
Electronic Location ID: e10484
Received 2020 Aug 28; Accepted 2020 Nov 12
Copyright: © 2020 Rouhan
Copyright year: 2020
Copyright holder: Rouhan
License: This is an open access article distributed under the terms of the Creative Commons Attribution License, which permits unrestricted use, distribution, reproduction and adaptation in any medium and for any purpose provided that it is properly attributed. For attribution, the original author(s), title, publication source (PeerJ) and either DOI or URL of the article must be cited.
License URL: https://creativecommons.org/licenses/by/4.0/

Keywords: Ferns, Identification keys, Madagascar, Western Indian Ocean, Morphological characters, New species, Scales, Polypodiopsida, Polypodiineae, Pteridophytes

Funding: MNHN This work was funded by MNHN, and in particular by ISYEB, ATM ‘Emergence des clades, des biotes et des cultures’ and ATM ‘Taxonomie moléculaire, DNA Barcode & gestion durable des collections’. The MNHN gives access to the collections in the framework of the RECOLNAT national Research Infrastructure. The funders had no role in study design, data collection and analysis, decision to publish, or preparation of the manuscript.

==============================
After 15 years of field studies in Madagascar, especially focused on the overlooked fern genus Elaphoglossum (Dryopteridaceae), a synoptic revision of the genus is here presented. Based on more than 2,600 herbarium specimens including collections over 200 years, Elaphoglossum is the second most diversified fern genus in Madagascar, with 52 species and three subspecies (with 76% of endemism). It is to be compared to the 34 species treated by Tardieu-Blot in 1960 for the “Flore de Madagascar et des Comores” or the 38 species listed by Roux in 2009 in the seminal “Synopsis of the Lycopodiophyta and Pteridophyta of Africa, Madagascar and neighboring islands”. The 55 taxa represent five out of seven existing generic sections (sect. Amygdalifolia and sect. Wrightiana being monotypic and Neotropical): sect. Lepidoglossa (29 spp. and three subspp.), sect. Elaphoglossum (17 spp.), sect. Setosa (3 spp.), sect. Squamipedia (2 spp.), and sect. Polytrichia (1 sp.). Distribution is given for each species and subspecies, and detailed for each island or archipelago in the Western Indian Ocean (La Réunion, Mauritius, Seychelles, and Comoros). Twenty species and three subspecies are newly described, all but one endemic to Madagascar: Elaphoglossum ambrense Rouhan, Elaphoglossum andohahelense Rouhan, Elaphoglossum anjanaharibense Rouhan, Elaphoglossum approximatum Rouhan, Elaphoglossum brachymischum Rouhan, Elaphoglossum cerussatum Tardieu subsp. brunneum Rouhan, Elaphoglossum coracinolepis Rouhan, Elaphoglossum desireanum Rouhan, Elaphoglossum glabricaule Rouhan, Elaphoglossum gladiifolium Rouhan, Elaphoglossum leucolepis (Baker) Krajina ex Tardieu subsp. nanolepis Rouhan, Elaphoglossum leucolepis (Baker) Krajina ex Tardieu subsp. nigricans Rouhan, Elaphoglossum longiacuminatum Rouhan, Elaphoglossum patriceanum Rouhan, Elaphoglossum perangustum Rouhan, Elaphoglossum prominentinervulum Rouhan, Elaphoglossum rakotondrainibeae Rouhan, Elaphoglossum repandum Rouhan, Elaphoglossum sabineanum Rouhan, Elaphoglossum sinensiumbrarum Rouhan, Elaphoglossum subglabricaule Rouhan, Elaphoglossum tsaratananense Rouhan, and Elaphoglossum viridicaule Rouhan. Morphological description, distribution map, and original illustrations are provided for each new taxon. Novel identification keys to the sections and all species from Madagascar are also presented.

Introduction

“Elaphoglossum is more in need of a taxonomical revision than other fern genera, all the more so as many species are imperfectly known or badly delimited in comparison with their allies”. Pichi Sermolli (1968)

“…Elaphoglossum is overlooked by seasoned collectors because of the monotony of their aspect for most of them and are therefore overlooked because they are just not different to the naked eye. So we can consider that a large number remains unknown”. Christ (1899)

Pichi-Sermolli’s opinion resulted from his studies of the genus Elaphoglossum in Africa “only”, and that of Christ is about the genus as a whole but over a century old. However, those two statements are still relevant today throughout most of the range of this fascinating fern genus growing primarily as epiphytes in all wet tropical forests.

Christ (1899) is the only one who undertook to monograph the genus worldwide with the 142 species he recognized at that time. Since then, field investigations, the study of herbarium collections, and progresses in taxonomy have sharply revised upwards these initial estimates, as it is commonly known that Elaphoglossum is one of the most diversified fern genera with 1,100 described species (IPNI, 2020) representing 586, 588, 600, 700, 730, 741, or 774 accepted species, according to respectively The Plant List (2013), The World Flora Online (WFO, 2020), PPGI (2016), Mickel & Atehortúa (1980), Hassler & Schmitt (2004–2020), Plants of the World Online (POWO, 2019), and The Leipzig Catalogue of Vascular Plants (Freiberg et al., 2020). Even so, the actual diversity could be much higher, given that the genus has been commonly overlooked and has been only revised entirely for some hotspots of biodiversity, in South Africa (Roux, 1982), in Peru (Mickel, 1991), Mesoamerica (Mickel, 1995a), Venezuela (Mickel, 1985, 1995b), Mexico (Mickel & Smith, 2004), Tropical East Africa (Mickel, 2002), Hawaii (Palmer, 2003), Mascarene Islands (Lorence & Rouhan, 2004; Lorence & Rouhan, 2008), French Polynesia (Rouhan et al., 2008), and Bolivia (Kessler et al., 2018). A few other recent taxonomic treatments are rather focused on small groups or sections in a given region (Matos & Mickel, 2014, 2018, 2019; Roux, 2011; Vasco, 2011; Vasco, Moran & Rouhan, 2009; Vasco, Mickel & Moran, 2013).

Elaphoglossum is monophyletic (Rouhan et al., 2004; Skog et al., 2004; Lóriga et al., 2014) and sister to Mickelia R.C.Moran, Labiak & Sundue, in the bolbitidoid clade (Moran, Labiak & Sundue, 2010) within the subfamily Elaphoglossoideae (Pic.Serm.) Crabbe of the Dryopteridaceae (Liu et al., 2016; PPGI, 2016). Although the genus is readily distinguished among the ferns, most of its species are morphologically uniform and difficult to distinguish: nearly all the species are indeed characterized by a transversely elongated ventral meristele in the rhizome, simple fronds (blades are divided in only four species; Vasco, Mickel & Moran, 2013), free veins, dimorphic sterile and fertile fronds, and acrostichoid sori (Rouhan et al., 2004; Moran et al., 2010). Contrasting with most fern genera, fertile fronds of Elaphoglossum are not critical for the taxonomy of the genus, and more generally, the number of useful characters is limited. Thus, the main characters used for describing species as well as intrageneric sections are the rhizome and frond scales. Mickel & Atehortúa (1980) proposed such a subdivision of the genus based on morphology and especially scales, and the six sections they defined have been largely supported by the molecular phylogenetic studies (Rouhan et al., 2004; Vasco et al., 2015; Vasco, Moran & Rouhan, 2009; Matos & Mickel, 2019), with a seventh section added by Lóriga et al. (2014). A study of spores in relation to phylogeny further supported many of the sections and subsections (Moran, Garrison Hanks & Rouhan, 2007).

Based on the currently known richness, Madagascar hosts just below 10% of the described species, but is, however, the second center of diversity for Elaphoglossum, behind the American tropics with ca. 75% of the described species. Madagascar is a continental island of the Western Indian Ocean known to host an extraordinary biodiversity, with 12,000–14,000 vascular plant species (Callmander, 2011; Lowry et al., 2018), and listed as one of the world’s hotspots of biodiversity (Myers et al., 2000). Eighty-three percent to 87% of vascular plants are endemics (Goodman & Benstead, 2005; Lowry et al., 2018), and among at least 600 species of ferns and lycophytes (Rakotondrainibe, 2003), with half probably endemic. All this diversity is however still underdocumented.

In the framework of the ongoing revision of all ferns and lycophytes of Madagascar, undertaken to update the “Flore de Madagascar et des Comores” treatment by Tardieu-blot (1960), I conducted the taxonomic revision of Elaphoglossum and identified 23 new species and subspecies for science. These new taxa are here fully described, illustrated, and mapped. The new species represent such an increase of the species diversity of the genus that I provide a novel dichotomous key including all Elaphoglossum species in Madagascar.

Materials and Methods

Field works

In addition to historical collections of Elaphoglossum over 200 years in Madagascar, collecting efforts especially focused on the genus were conducted in the Island since 2004, mainly in protected areas, as most, if not all, of the remaining wet natural forests are included in National Parks and other natural reserves. Collecting permits were granted by Madagascar National Parks and the Ministère de l’Environnement, et du Développement Durable (project numbers: 70/19/MEDD/SG/DGF/DSAP/SCB.Re, and 207/15/MEEMF/SG/DGF/DAPT/SCBT, and 199/15/MEEMF/SG/DGF/DAPT/SCBT, and 241/11/MEF/SG/DGF/DCB.SAP/SCB).

Plants were systematically sampled as modern collections, that is, including herbarium specimens, silica-dried leaf sample, and photos (Gaudeul & Rouhan, 2013). Complete sets of all collections made during these field trips are deposited at TAN or TEF, and with a few exceptions at P; duplicates, when available, have been sent elsewhere (or will be sent right away after publication) particularly to K, MO, NBG, NY (herbarium codes follow Thiers, 2018).

Herbarium-based studies

The taxonomic revision that led to defining taxa and building novel identification keys is based on the examination of over 2,600 herbarium specimens representing 2,186 gatherings housed at P, and on-field observations of most Elaphoglossum species. All specimens were databased and are freely available in the Paris Herbarium database at https://science.mnhn.fr/institution/mnhn/collection/p/item/search?lang=en_US. Additional specimens from other herbaria were examined in hand and annotated (BM, G, K, MO, NY, P, PR, TAN, TEF, US) or examined as online images (B, BR, PRE). All measurements, colors and other details included in the descriptions were based on herbarium specimens and data derived from field notes. In evaluating the variability of each species, habitat and ecology were noted in the field, but information on these features were also taken from other herbarium labels.

Illustrations and morphological characters

Herbarium specimens were examined under dissection microscope Leica MZ6, and close-up images acquired through a camera Leica DFC425 provided illustrations for each taxon; scales were mounted in glycerin gelatin between slide and slip-cover, and these permanent slides were imaged using a slide scanner Nikon CoolScan V ED. The terminology used to describe the plants is based on Lellinger (2002).

Distribution maps

Distribution maps of new taxa were based on all cited specimens and generated with QGIS 2.14 (QGIS Geographic Information System. Open Source Geospatial Foundation Project. http://qgis.osgeo.org). A background map included five altitudinal ranges corresponding globally to those generally recognized in Madagascar (Humbert, 1955; Faramalala, 1995): 0–400 m (green), 400–800 m (yellow), 800–1,200 m (light brown), 1,200–1,800 m (medium brown) and >1,800 m (dark brown). Localities of specimens were represented by red dots (and open circles represented the six main cities in Madagascar). Distribution is also described in the text for each species and subspecies according to the five Malagasy phytogeographic domains as defined by Humbert (1955): East, Sambirano, Center, West, and South.

New botanical taxa

New botanical taxa were described only after considering all species known at least in Madagascar, Africa, Western Indian Ocean Islands (Comoros, Seychelles, La Réunion, Mauritius), and circumaustral islands from the Atlantic and the Indian Ocean. Thus, a morphological comparison to most closely-related species from those areas is provided through diagnoses and keys.

The electronic version of this article in Portable Document Format (PDF) will represent a published work according to the International Code of Nomenclature for algae, fungi, and plants (ICN), and hence the new names contained in the electronic version are effectively published under that Code from the electronic edition alone. In addition, new names contained in this work which have been issued with identifiers by IPNI will eventually be made available to the Global Names Index. The IPNI LSIDs can be resolved and the associated information viewed through any standard web browser by appending the LSID contained in this publication to the prefix “http://ipni.org/”. The online version of this work is archived and available from the following digital repositories: PeerJ, PubMed Central, and CLOCKSS.

Results and Discussion

With 52 species and three subspecies, Elaphoglossum ranks as the second most diverse fern genus in Madagascar (after Asplenium L.) (Table 1). This species diversity is to be compared to the 34 species treated by Tardieu-Blot in the Flora of Madagascar (1960) or the 38 species more recently listed by Roux (2009). The 55 taxa take account of the identification of 20 new species for science, and three new subspecies, all but one endemic to Madagascar. New taxa were first discovered not only by intensive and focused prospections in the field, but also by in-depth studies of a plethora of unidentified and misidentified earlier specimens kept in Herbaria: in particular, the Paris herbarium (P) houses one of the most important collections of Malagasy plants (Le Bras et al., 2017), and likely the most diversified and abundant collection of the Malagasy ferns (more than 38,800 specimens), thanks to general collectors who devoted their lives to the flora of Madagascar (e.g., Henri Perrier de la Bâthie and Henri Humbert), to pteridologists who gathered critical and abundant collections made by others (Roland Bonaparte), and to more recent pteridologists like France Rakotondrainibe or myself.

Table 1 Species (52) and subspecies (3) of Elaphoglossum recognized in Madagascar, endemism status, and occurrence elsewhere with all Islands of Western Indian Ocean detailed.

Section	Species in Madagascar	Madagascar	Comoros	La Réunion	Mauritius	Seychelles	Africa	Others	
Lepidoglossa	Elaphoglossum achroalepis (Baker) C.Chr.	endemic							
Elaphoglossum	Elaphoglossum acrostichoides (Hook. & Grev.) Schelpe	native	native	native			native		
Lepidoglossa	Elaphoglossum ambrense Rouhan, sp. nov.	endemic							
Lepidoglossa	Elaphoglossum andohahelense, Rouhan sp. nov.	endemic							
Elaphoglossum	Elaphoglossum angulatum (Blume) T.Moore	native		native			native	Asia, Oceania	
Lepidoglossa	Elaphoglossum anjanaharibense Rouhan, sp. nov.	endemic							
Elaphoglossum	Elaphoglossum approximatum Rouhan, sp. nov.	endemic							
Lepidoglossa	Elaphoglossum aspidiolepis (Baker) C.Chr.	endemic							
Lepidoglossa	Elaphoglossum asterolepis (Baker) C.Chr.	endemic							
Setosa	Elaphoglossum aubertii (Desv.) T.Moore	native	native	native			native		
Lepidoglossa	Elaphoglossum avaratraense Rakotondr.	endemic							
Lepidoglossa	Elaphoglossum brachymischum Rouhan, sp. nov.	endemic							
Lepidoglossa	Elaphoglossum capuronii Tardieu	endemic							
Lepidoglossa	Elaphoglossum cerussatum Tardieu subsp. brunneum, Rouhan, subsp. nov.	endemic							
Lepidoglossa	Elaphoglossum cerussatum Tardieu, subsp. cerussatum	endemic							
Lepidoglossa	Elaphoglossum conforme (Sw.) J.Sm.	native					native	St Helena	
Lepidoglossa	Elaphoglossum coracinolepis Rouhan, sp. nov.	endemic							
Elaphoglossum	Elaphoglossum coriaceum Bonap.	native		native	native		native		
Elaphoglossum	Elaphoglossum coursii Tardieu	native	native	native			native		
Elaphoglossum	Elaphoglossum decaryanum Tardieu	endemic							
Lepidoglossa	Elaphoglossum desireanum Rouhan, sp. nov.	endemic							
Lepidoglossa	Elaphoglossum forsythii-majoris Christ	endemic							
Lepidoglossa	Elaphoglossum glabricaule Rouhan, sp. nov.	endemic							
Lepidoglossa	Elaphoglossum gladiifolium Rouhan, sp. nov.	endemic							
Lepidoglossa	Elaphoglossum humbertii C.Chr.	endemic							
Polytrichia	Elaphoglossum hybridum (Bory) Brack.	native	native	native	native		native	Tristan da Cunha, Gough Island, Inaccessible Island, South America	
Lepidoglossa	Elaphoglossum lancifolium (Desv.) C.V.Morton	native	native	native	native	native	native		
Elaphoglossum	Elaphoglossum lepervanchei (Bory ex Fée) T.Moore	native	native	native	native	native	native		
Lepidoglossa	Elaphoglossum leucolepis (Baker) Krajina ex Tardieu subsp. leucolepis	endemic							
Lepidoglossa	Elaphoglossum leucolepis (Baker) Krajina ex Tardieu subsp. nanolepis Rouhan, subsp. nov.	endemic							
Lepidoglossa	Elaphoglossum leucolepis (Baker) Krajina ex Tardieu subsp. nigricans Rouhan, subsp. nov.	endemic							
Elaphoglossum	Elaphoglossum longiacuminatum Rouhan, sp. nov.	endemic							
Elaphoglossum	Elaphoglossum malgassicum C.Chr.	endemic							
Squamipedia	Elaphoglossum marojejyense Tardieu	endemic							
Lepidoglossa	Elaphoglossum multisquamosum Bonap.	native		native					
Squamipedia	Elaphoglossum nidusoides Rouhan & Rakotondr.	endemic							
Elaphoglossum	Elaphoglossum ovalilimbatum Bonap.	endemic							
Lepidoglossa	Elaphoglossum patriceanum Rouhan, sp. nov.	endemic							
Elaphoglossum	Elaphoglossum perangustum Rouhan, sp. nov.	endemic							
Lepidoglossa	Elaphoglossum perrierianum C.Chr.	endemic							
Setosa	Elaphoglossum phanerophlebium C.Chr.	native					native		
Lepidoglossa	Elaphoglossum poolii (Baker) Christ	endemic							
Elaphoglossum	Elaphoglossum prominentinervulum Rouhan, sp. nov.	endemic							
Lepidoglossa	Elaphoglossum pseudovillosum Bonap.	endemic							
Elaphoglossum	Elaphoglossum rakotondrainibeae Rouhan, sp. nov.	endemic							
Lepidoglossa	Elaphoglossum repandum Rouhan, sp. nov.	endemic							
Lepidoglossa	Elaphoglossum rufidulum (Willd. ex Kuhn) C.Chr.	endemic							
Elaphoglossum	Elaphoglossum sabineanum Rouhan, sp. nov.	endemic							
Lepidoglossa	Elaphoglossum scolopendriforme Tardieu	endemic							
Elaphoglossum	Elaphoglossum sinensiumbrarum Rouhan, sp. nov.	endemic							
Setosa	Elaphoglossum spatulatum (Bory) T.Moore	native		native	extinct? (not collected for over a century)		native	Sri Lanka	
Lepidoglossa	Elaphoglossum subglabricaule Rouhan, sp. nov.	endemic							
Elaphoglossum	Elaphoglossum subsessile (Baker) C.Chr.	endemic							
Lepidoglossa	Elaphoglossum tsaratananense Rouhan, sp. nov.	endemic							
Elaphoglossum	Elaphoglossum viridicaule Rouhan, sp. nov.	native	native						

The endemism in the genus is high in Madagascar (76% representing 39 species and three subspecies), with five sections represented. The two remaining sections, sect. Amygdalifolia and sect. Wrightiana, are monotypic and endemic to the Neotropics (Mickel & Atehortúa, 1980; Lóriga et al., 2014). This species diversity in Madagascar was best explained by more than 10 independent long distance dispersal events from the Neotropics to Madagascar giving rise to many distinct lineages followed by insular speciations (Rouhan et al., 2004).

With 29 species, sect. Lepidoglossa is the most diversified in Madagascar, followed by sect. Elaphoglossum (17 spp.), reflecting their likely most important diversities also at the world level (Table 2).

Table 2 Taxonomic diversity and endemism of the five sections of Elaphoglossum represented in Madagascar (sect. Amygdalifolia and Wrightiana not represented).

Sections	Numbers of species (endemic)	Number of subspecies (endemic)	Endemism (all taxa)	
I : sect. Polytrichia	1 (0)	–	0%	
II : sect. Setosa	3 (0)	–	0%	
III : sect. Elaphoglossum	17 (11)	–	64,7%	
IV : sect. Squamipedia	2 (2)	–	100%	
V : sect. Lepidoglossa	29 (26)	3 (3)	90,6%	
Total	52 (39)	3 (3)	76,4%	

Section Polytrichia and Setosa, together forming the subulate scales clade, are poorly represented by a few non-endemic species: Elaphoglossum hybridum (Bory) Brack. is even unique in the genus in being widespread from South America to Africa, Madagascar and nearby islands (Matos & Mickel, 2014).

By contrast, only two species in section Squamipedia are endemic to Madagascar, and all other 16 species of the section are endemic to the Neotropics (Vasco, Mickel & Moran, 2013; Vasco et al., 2015). The two Malagasy species, E. nidusoides Rouhan & Rakotondr. and E. marojejyense Tardieu, are sister group to the Neotropical species and are morphologically atypical in sect. Squamipedia (Rouhan, Rakotondrainibe & Moran, 2007; Vasco et al., 2015). As echinulate perine ornementations are a synapomorphy for the Neotropical clade (Moran, Garrison Hanks & Rouhan, 2007; Vasco et al., 2015), the two Malagasy species may eventually end up in a new section considering especially their unusal and unique succulent-like blades.

Taxonomic Results

Elaphoglossum Schott ex J.Sm., J. Bot. (Hooker) 4 (27): 148 (1841), nom. et typ. cons. Type: Elaphoglossum conforme (Sw.) J.Sm.; Acrostichum conforme Sw. (type designated by J.Sm., Hist. Fil.: 125 (1875)).

Key to the sections of Elaphoglossum in Madagascar

Note: diameter measurements are given without scales for rhizomes and petioles. Stomata, sometimes numerous and colorful, should not be confused with glutinous dots. Since no morphological character allow distinguishing reliably section Elaphoglossum from sect. Squamipedia in Madagascar (Rouhan, Rakotondrainibe & Moran, 2007; Matos, Vasco & Moran, 2018), the two sections are grouped in a single couplet.

 1a. Laminae without scales or glutinous dots, but with scales that are inconspicuous, reduced (<0.5 mm in diam.), arachnidoid, scattered, and dark brown to black; rhizomes non-glutinousIII and IV. sect. Elaphoglossum/sect. Squamipedia

 1b. Laminae either with scales scattered to dense, of various sizes, colors, and shapes but non-arachnidoid, or without scales but then usually with glutinous dots; rhizomes glutinous or non-glutinous.

  2a. Frond scales subulate (i.e., hairlike or slender conical appearance: patent, with strongly involute margins from a wider base) and patent; laminae without glutinous dots.

   3a. Hydathodes presentII. sect. Setosa

   3b. Hydathodes absentI. sect. Polytrichia

  2b. Frond scales, when present, non-subulate, flat or hemi-infundibular at base; when scales are absent, laminae usually with glutinous dotsV. sect. Lepidoglossa

Key to the species and subspecies of Elaphoglossum in Madagascar

Section I – Sect. Polytrichia (Sodiro) Christ: taxa with patent, subulate (inrolled) scales on fronds; hydathodes absent in adult fronds; frond scales dark brown to black, denser on petioles, laminar margins and median veins. One species: E. hybridum.

Section II – sect. Setosa (Christ) Mickel & Atehortúa: taxa with patent, subulate (inrolled) scales on fronds; hydathodes present in adult fronds (if inconspicuous, the sterile laminae <11 cm long and spathulate to oblanceolate, and fertile laminae conduplicate before maturity); frond scales light brown or orange brown.

1a. Sterile laminae 8–40 cm long, narrowly oblong to linear; epiphytic or epilithicE. aubertii

1b. Sterile laminae (0.8–)1.5–7.0(–9.0) cm long, oblong, oblanceolate, spathulate, or elliptic; epilithic on rocks of riverbeds or steep banks (rarely epiphytic).

 2a. Sterile laminae elliptic to oblong, usually membranaceous almost translucent (sometimes herbaceous) with distinct veins on both surfaces; fertile laminae non-conduplicate before maturityE. phanerophlebium

 2b. Sterile laminae spathulate to oblanceolate, chartaceous to coriaceous, veins little or not distinct on both surfaces; fertile laminae conduplicate before maturityE. spatulatum

Sections III and IV – Sect. Elaphoglossum and sect. Squamipedia Mickel & Atehortúa: species with rhizomes non-glutinous, scaly; rhizome scales entire or ciliate with non-acicular cilia; laminae glabrous or most often with inconspicuous, reduced arachnidoid scales, fronds without glutinous dots.

Since no morphological character allow distinguishing one section from the other in Madagascar (see Rouhan, Rakotondrainibe & Moran, 2007; Matos, Vasco & Moran, 2018), the two sections are included in a single key.

1a. Rhizome scales mostly coriaceous to sclerotic and opaque, concolorous, black or dark castaneous, bright, usually 0.3–2.5 mm long.

 2a. Rhizomes 0.5–2.5 mm diam.

  3a. Sterile laminae linear to narrowly elliptic, with apices attenuate to angustate; median veins immersed at a level below the adaxial surface of the lamina (hidden in a longitudinal sulcus); rhizome scales densely distributed to somewhat scattered.

   4a. Fronds 1.5–3.0 mm apart, subsessile; rhizome scales 0.5–1.5(–2.0) mm long; fertile fronds shorter than the sterile ones; epiphyticE. coriaceum

   4b. Fronds (3–)5–20 mm apart, petiolate; rhizome scales (0.5–)1.7–2.5(–3.0) mm long; fertile fronds slightly longer than, or as long as the sterile ones; epiphytic or terrestriaE. coursii

  3b. Sterile laminae ovate, lanceolate, spathulate, or shortly oblong, with apices round, obtuse or acute; median veins complanate, or prominulous and slightly sulcate on the adaxial surface; rhizome scales very scattered.

  5a. Laminae carnose, succulent-like; median veins non apparentE. marojejyense (sect. Squamipedia).

  5b. Laminae coriaceous; median veins apparent.

   6a. Sterile laminae ovate or shortly oblong, with bases overall obtuse and shortly acuminate at the end; median veins complanate on the adaxial surfaceE. ovalilimbatum

   6b. Sterile laminae lanceolate, with bases acuminate and long-decurrent; median veins prominulous and slightly sulcate on the adaxial surfaceE. sabineanum, sp. nov.

2b. Rhizomes 2.5–13.0 mm diam.

 7a. Sterile fronds sessile or subsessile (petioles <3/20 of total frond length); sterile laminae narrowly oblong to oblanceolate; rhizome scales dark castaneous to ferrugineousE. malgassicum

 7b. Sterile fronds petiolate (petiole >¼ of total frond length); sterile laminae elliptic; rhizome scales black or dark castaneous sometimes with blackish spots.

  8a. Lateral veins conspicuous, prominulous and tangible on both surfaces; rhizome apices pale yellowish (in natura)……E. prominentinervulum, sp. nov.

  8b. Lateral veins inconspicuous on both surfaces; rhizome apices green (in natura)E. viridicaule, sp. nov.

1b. Rhizome scales mostly papyraceous to scarious, more or less translucent (at least at apex of rhizomes), concolorous or bicolorous, of various colors (stramineous, orange brown, brown, dark castaneous to black), bright or matte, 2.5–15.0 mm long.

 9a. Rhizome scales black to dark castaneous, bright (at least in part).

  10a. Sterile laminae narrowly linearE. perangustum, sp. nov.

  10b. Sterile laminae elliptic, oblong or ovate

   11a. Rhizome scales concolorous (dark castaneous) or bicolorous (castaneous/black, or light brown/black), ovate to lanceolate; rhizome apices pale yellowish (in natura).

    12a. Lateral veins inconspicuous on both surfacesE. acrostichoides

    12b. Lateral veins conspicuous, prominulous and tangible on both surfacesE. prominentinervulum, sp. nov.

   11b. Rhizome scales concolorous (black), narrowly lanceate to lanceolate; rhizome apices green (in natura).

    13a. Rhizomes short-creeping, fronds 3–5(–7) mm apartE. approximatum, sp. nov.

    13b. Rhizomes long-creeping, fronds (5–)6–30(–60) mm apart.

     14a. Sterile laminar bases long-decurrent in a narrow wing bordering the petiole, apices round to obtuse; rhizomes 3.2–5.0 mm diamE. longiacuminatum, sp. nov.

     14b. Sterile laminar bases short-acuminate, apices attenuate; rhizomes 1.2–2.5(–3.5) mm diamE. rakotondrainibeae, sp. nov.

9b. Rhizome scales stramineous, light brown or orange brown, matte.

 15a. Rhizome scales ovate, 2–3 mm broad, tightly appressed against rhizomes.

  16a. Rhizomes long-creeping with fronds distant, 10–40 mm apart; laminae coriaceous; rhizomes 2–3 mm diamE. angulatum

  16b. Rhizomes short-creeping with fronds clustered, 2–5 mm apart; laminae carnose, more or less succulent-like; rhizomes 4–8 mm diamE. nidusoides (sect. Squamipedia)

 15b. Rhizome scales linear to lanceolate, 0.3–2.0 mm broad, not appressed against rhizomes.

  17a. Rhizomes erect like a small caudex; terrestrial (exceptionally epiphytic)E. subsessile

  17b. Rhizomes short-creeping; terrestrial or epiphytic.

   18a. Rhizome scales with numerous marginal cilia, non-glandular, contorted and intricate; rhizomes 4–10 mm diamE. decaryanum

   18b. Rhizome scales with some marginal cilia, glandular, and contorted; rhizomes 2.5–5.0 mm diam.

    19a. Lateral veins inconspicuous on both surfaces; laminae of various shapesE. lepervanchei

    19b. Lateral veins conspicuous, prominulous and tangible on both surfaces; laminae narrowly ellipticE. sinensiumbrarum, sp. nov.

Section V – Sect. Lepidoglossa Christ: taxa with laminae scaly or without scales but with glutinous dots; frond scales flat or with a hemi-infundibular base; rhizomes scaly or glabrous. Glutinous dots are more or less spread, at first often translucent and bright, colorless to ferrugineous, turning opaque, whitish or black. Laminar scales are sometimes glabrescent and inconspicuous, but they are numerous and never reduced and arachnidoid as in section Elaphoglossum. Elaphoglossum acrostichoides from the latter section is however here included as it can be quite unusual in having many scales on laminae, resulting in potential confusions between the two sections.

1a. Rhizomes glabrous or scaly; rhizome scales (when present) dark, concolorous (black, dark castaneous, or dark ferrugineous), mostly bright and brittle.

 2a. Rhizomes glabrous (or subglabrous) and glutinous.

  3a. Sterile laminae linear (5–35 mm broad), narrowly elliptic or oblanceolate, without scales but with numerous glutinous dots.

   4a. Rhizomes long-creeping (fronds 10–30 mm apart), glabrous; laminae linear to narrowly elliptic, apices cuneateE. humbertii

   4b. Rhizomes short-creeping (fronds <5 mm apart), with rare, scattered scales; laminae oblanceolate, apices obtuse to roundE. subglabricaule, sp. nov.

  3b. Sterile laminae linear (1–6 mm broad), glabrous or densely scaly on the abaxial surface but without glutinous dots.

   5a. Sterile laminae glabrous on both surfaces, 1.0–2.3 mm broad; rhizomes long-creeping (fronds 3–5 mm apart), glabrousE. glabricaule, sp. nov.

   5b. Sterile laminae densely scaly on the abaxial surface, 3–6 mm broad; rhizomes short-creeping (fronds <5 mm apart), rare scales towards the apexE. perrierianum

 2b. Rhizomes scaly (scales conspicuous, scattered to dense), glutinous or not.

 6a. Sterile laminae with glutinous dots on both surfaces; dots orange brown to black, sometimes turning whitish; laminar scales present or absent.

  7a. Laminar scales stellate (body suborbicular, with marginal cilia as rays).

   8a. Rhizome scales 2.5–3.5 × 0.3–0.5 mm, margins entire (non-ciliate); abaxial laminar scales stellate with usually >8 marginal cilia (rays), shorter than, or as long as the suborbicular body; large populations, epilithic on riverbeds and river banks; AnjanaharibeE. anjanaharibense, sp. nov.

   8b. Rhizome scales 3–5 × 0.5–0.8 mm, margins with few acicular cilia in the basal half; abaxial laminar scales stellate with usually 6–8 acicular cilia longer than the scale body reduced to the point of attachment; epiphytic or epilithic; from North to SouthE. lancifolium

  7b. Laminar scales absent, or when present, ovate, lanceate, or lanceolate.

   9a. Laminae wihout scalesE. gladiifolium, sp. nov.

   9b. Laminae with scales (sometimes inconspicuous but persistent at least on margins and/or along median veins).

    10a. Laminae soft and herbaceous

    11a. Rhizome scales more or less patent (2.5–)3.0–5.5 mm long; sterile laminar bases narrowly cuneate, apices cuneate to acuteE. ambrense, sp. nov.

    11b. Rhizome scales more or less appressed, antrorse, 1.5–2.5 mm long; sterile laminar bases attenuate, long-decurrent, apices angustateE. andohahelense, sp. nov.

    10b. Laminae rigid and coriaceousE. patriceanum, sp. nov.

6b. Sterile laminae without dots; laminar scales present.

 12a. Rhizome scales entire or with marginal cilia; cilia glandular, contorted, <0.5 mm long (sometimes with some acicular cilia as well).

  13a. Rhizomes short-creeping, 3.2–5.0 mm diam.; laminae elliptic, 1.0–2.5 cm broad, apices acute to acuminate; Tsaratanana (North).

   14a. Petiole scales 0.4–0.6 mm broad, lanceate, bases cordate or truncate, apices long-acuminate, margins entire above the hastate baseE. pseudovillosum

   14b. Petiole scales 0.7–1.5 mm broad, lanceate, bases hastate, apices cuneate, margins ciliate with 15–30 pairs of acicular cilia (0.2–0.5 mm long)E. tsaratananense, sp. nov.

  13b. Rhizomes long-creeping, 0.9–1.7 mm diam.; laminae narrowly oblanceolate to oblong, 0.5–1.0 cm broad, apices obtuse to round; SouthE. repandum, sp. nov.

 12b. Rhizome scales with marginal cilia; cilia straight and acicular, either numerous (5–30 pairs) and short (<0.5 mm), or few (<10 pairs) and long (>0.5 mm).

  15a. Sterile laminae 2–5 mm broad, linear, margins partly or entirely revolute.

    16a. Laminar scales linear, 1.5–2.0 mm long, margins with 3–7 pairs of long cilia; rhizome scales narrowly lanceolate to linear, marginal cilia acicular and shorter than the scale bodyE. forsythii-majoris

    16b. Laminar scales orbicular, reduced to the point of attachment plus 1–4 acicular cilia, 0.5–2.0(–3.0) mm long; rhizome scales suborbicular, marginal cilia acicular and longer than the scale bodyE. avaratraense

  15b. Sterile laminae 7–30 mm broad, narrowly obovate, oblong, lanceolate, or elliptic, margins not revolute.

   17a. Petiole scales more or less bicolorous: dark at base and center, light at margins (or sometimes dark), and lighter marginal cilia.

    18a. Laminar scales dense, imbricate, and regularly arranged as fish scales, 3–5 × 0.8–1.5 mm, with 30–50 pairs of marginal cilia.

     19a. Petiole scales clearly bicolorous (central area of apex and margins white to light orange brown, matte; central area of base dark castaneous), sometimes also present in the basal half of abaxial median veins; apex of sterile laminae obtuse to round; North and CenterE. leucolepis subsp. leucolepis

     19b. Petiole scales largely concolorous, black to dark brown (marginal cilia, at least, are lighter), covering also at least the basal half of abaxial median veins; apex of sterile laminae obtuse to acute; Center and SouthE. leucolepis subsp. nigricans, subsp. nov.

    18b. Laminar scales more or less dense, sometimes overlapping but irregularly and leaving spaces; scales 1.0–3.0 × 0.3–0.8 mm with <30 pairs of marginal ciliaE. leucolepis subsp. nanolepis, subsp. nov.

   17b. Petiole scales bicolorous: light at center, dark at margins, and dark marginal cilia.

    20a. Laminar scales non-imbricate, 0.3–1.0(–1.4) × 0.1–0.3 mm, with 4–8 pairs of marginal cilia up to eight times the width of the scale bodyE. asterolepis

    20b. Laminar scales imbricate (obscuring usually at least the abaxial laminae), 1.5–2.5 × 0.4–1.0 mm, with 8–20 pairs of marginal cilia up to twice the width of the scale bodyE. rufidulum

1b. Rhizomes scaly; rhizome scales light colored, concolorous or bicolorous (whitish, orange brown, castaneous or brown), matte or bright, mostly papyraceous to scarious.

 21a. Sterile laminae with, on both surfaces, dots orange brown to black.

  22a. Rhizomes long-creeping.

   23a. Rhizome scales ovate to lanceolate, 5–10 × 1–2 mmE. conforme

   23b. Rhizome scales narrowly lanceate to linear, 1.5–5.5 ×0.2–0.5 mm.

    24a. Laminar scales stellate (body suborbicular, with marginal cilia as rays); large populations, epilithicE. anjanaharibense, sp. nov.

    24b. Laminar scales ovate to lanceolate (sometimes inconspicuous but persistent at least on margins and/or along median veins); isolated individuals, epiphytic.

     25a. Rhizome scales more or less patent (2.5–)3.0–5.5 × 0.3–0.5 mm; sterile laminar bases cuneate, apices cuneate to acuteE. ambrense, sp. nov.

     25b. Rhizome scales more or less appressed, antrorse, 1.5–2.5 × 0.2–0.4 mm; sterile laminar bases attenuate, long-decurrent, apices angustateE. andohahelense, sp. nov.

 22b. Rhizomes short-creeping.

  26a. Sterile fronds 10–35 cm long, petioles <⅓ of total frond length; sterile laminar bases attenuate, apices obtuse to roundE. aspidiolepis

  26b. Sterile fronds 35–70 cm long, petioles >½ of total frond length; sterile laminar bases and apices cuneateE. capuronii

21b. Sterile laminae without dots.

 27a. Vein apices anastomosed into an intramarginal vein, continuous or discontinuous; rhizome scales concolorous, white, castaneous, or brown.

  28a. Sterile laminae oblong (rarely obovate), bases cuneate to obtuse, apices round or obtuse (rarely acute).

   29a. Laminar scales blackE. coracinolepis, sp. nov.

   29b. Laminar scales white or light orange brown

    30a. Rhizome scales white (rarely light orange brown), matteE. cerussatum subsp. cerussatum

    30b. Rhizome scales castaneous, matte but sometimes slightly brightE. cerussatum subsp. brunneum, subsp. nov.

  28b. Sterile laminae lanceate, lanceolate, or elliptic, bases cuneate to acuminate, apices cuneate, attenuate or acuminate.

   31a. Rhizome scales white, with numerous marginal cilia; cilia contorted, 2–3(–4) × 0.3–1.0 mm; intramarginal veins continuousE. achroalepis

   31b. Rhizome scales light brown, subentire, 4.0–6.5 × 1.0–1.8 mm; intramarginal veins discontinuous (sometimes absent)E. desireanum, sp. nov.

  27b. Vein apices free (without intramarginal vein); rhizome scales concolorous or bicolorous, light brown to castaneous.

   32a. Rhizome scales linear to lanceate, 0.2–0.8 mm broad.

    33a. Laminae 4.5–8.0 × 0.5–1.0 cm, apices obtuse to round; rhizomes long-creeping, 0.9–1.7 mm diam.E. repandum, sp. nov.

    33b. Laminae 7.3–18 × 1.3–2.5 cm, apices acuminate; rhizomes short-creeping, 3.3–4.0 mm diam.

    34a. Sterile fronds subsessile or short-petiolate (<1/5 of total frond length), with scales truncate or cordate at base; rhizome scales 1.5–2.5 × 0.2–0.4 mmE. brachymischum, sp. nov.

    34b. Sterile fronds petiolate (>2/5 of total frond length) with scales hastate at base; rhizome scales 2.5–4.5 × 0.5–0.8 mm.E. pseudovillosum

  32b. Rhizome scales ovate to lanceolate, 1.0–2.3 mm broad.

   35a. Rhizome scales somewhat bicolorous (dark areas alternate with light ones) E. acrostichoides (sect. Elaphoglossum)

   35b. Rhizome scales concolorous.

   36a. Petiole scales rare to scattered, with non-ciliate margins; laminar scales irregularly-shaped, lanceolate to substellate, scattered, 0.3–1.0 × 0.2–0.5 mm.

    37a. Rhizome scales 4.0–6.5 mm long; lateral veins 1–2 mm apart, and at 40°–50° angle to median veinE. desireanum, sp. nov.

    37b. Rhizome scales 1.5–3.5 mm long; lateral veins (1.5–)2.0–4.0 mm apart, and at 50°–70° angle to median veinE. scolopendriforme

   36b. Petiole scales dense, with marginal cilia; laminar scales ovate to lanceolate, dense, 1.5–4.0 × 0.5–1.5 mm.

    38a. Fronds narrowly elliptic or oblanceolate (rarely linear); sterile petioles <½ of total frond length; laminar scales ciliate (10–15(20) pairs of acicular cilia, often as long as the width of the scale body)E. multisquamosum

    38b. Fronds oblong; sterile petioles >½ of total frond length; laminar scales ciliate (10)20–40 pairs of acicular cilia shorter than the width of the scale bodyE. poolii

Descriptions of the 20 new species and 3 new subspecies

Elaphoglossum ambrense Rouhan, sp. nov. (Figs. 1 and 2)

Figure 1 Elaphoglossum ambrense Rouhan.

(A) Habit. (B) Rhizome. (C) Rhizome scale. (D) Laminar scale. (E) Detail of sterile lamina, abaxial surface. (F) Habit in natura. A–E, Rouhan 311, P00749250. F, Rouhan 304, P00749243. Photos: G. Rouhan/MNHN.

Figure 2 Distribution map of Elaphoglossum ambrense.

Red dots represent localities of specimens, and open circles represent the six main cities in Madagascar; the five altitudinal ranges corresponding globally to those generally recognized in Madagascar (Humbert, 1955; Faramalala, 1995) are represented in green (0–400 m), yellow (400–800 m), light brown (800–1,200 m), medium brown (1,200–1,800 m), and dark brown (>1,800 m).

Type:–MADAGASCAR. Région de Diana, District d’Antsiranana II, Parc national Montagne d’Ambre, bord de piste reliant la station des Roussettes au Grand Lac, entre Grand Lac et piste menant au Lac Maudit, 1,300 m, 12°35′32″S 49°9′20″E, 9 Oct. 2004, G. Rouhan, T. Janssen, H.L. Ranarijoana, E. Randrianjohany 311 (holotype: P [P00749250!]; isotypes: NBG, NY, P [P00749251!], TEF).

Diagnosis:–Elaphoglossum ambrense differs from the much more widespread E. lancifolium (Desv.) C.V.Morton by the absence of stellate scales on the fronds, and by entire, more flexible, and non-ferrugineous rhizome scales. The rhizome scales of E. ambrense and E. anjanaharibense Rouhan are similar, but E. ambrense differs in the absence of stellate scales on the fronds, broader (>1 cm) and non-angustate laminae, fertile fronds longer than the sterile ones, the epiphytic habitat, and plants not forming large populations. The rhizome scales are also somewhat similar to those of E. andohahelense Rouhan but they are more or less patent (vs. appressed) on thicker rhizomes, and the laminar scales are shorter to ovate, and distributions of the two species do not overlap.

Description:–Rhizomes long-creeping, 2.0–3.4 mm diam., branched, glutinous, densely scaly, especially towards apices (Fig. 1B); rhizome scales more or less patent and divergent, narrowly lanceate to linear (2.5–)3.0–5.5 × 0.3–0.5 mm, concolorous, castaneous to dark castaneous, bright especially at the apex, translucent, scarious to papyraceous, flexible, basifixed, bases truncate to cordate, apices long-attenuate to angustate, margins entire (Fig. 1C). Sterile fronds erect, inserted in two separate rows, 5–13 mm apart, 15–34 cm long (Fig. 1A); petioles 4–10 cm long, 0.9–1.5 mm diam., moderately and deciduously scaly; petiole scales non-subulate, similar to the rhizome scales but shorter, to 2.5 mm long, sometimes mixed with other scales, irregularly shaped, roughly oval, to 0.5 mm long; sterile laminae herbaceous, narrowly elliptical to linear, 10–24 × 1.0–1.5 cm, bases narrowly cuneate, apices cuneate to acute, margins undulate, both surfaces deciduously scaly and with rare to dense glutinous dots, red to black (persistent points of attachment of scales) (Fig. 1E); laminar scales scattered, persisting longer on and along median veins, non-subulate, irregularly shaped, ovate to lanceolate, 0.3–1.0 × 0.1–0.2 mm, castaneous to light red, peltate, margins subentire to erose (Fig. 1D); median veins prominulous on both surfaces, rounded or slightly sulcate on abaxial surface, slightly sulcate on adaxial surface, scales similar to those of laminae; lateral veins visible, free, simple or bifurcate, apices enlarged and darker without hydathode.

Fertile fronds longer than the sterile ones, longer petiolate (3/10– ½ of total frond length), laminae linear.

Etymology:–The specific epithet derives from the ‘Montagne d’Ambre’ National Park, a mountain area rich in ferns, and the single locality known for the species.

Habitat and distribution:–Endemic to Madagascar, Elaphoglossum ambrense is rare and grows as epiphytic in wet evergreen forests of Montagne d’Ambre (North), in the Central phytogeographic domain, 1,300–1,475 m (Fig. 2).

Additional specimens examined (paratypes):–MADAGASCAR. Malcomber 2350 (P). - Nusbaumer 2403 (P). - Rouhan 304 (P)

Elaphoglossum andohahelense Rouhan, sp. nov. (Figs. 3 and 4)

Figure 3 Elaphoglossum andohahelense Rouhan.

(A) Habit. (B) Rhizome. (C) Rhizome apex. (D) Rhizome scale. (E) Laminar scale. (F) Detail of sterile lamina, abaxial surface. A–F, Rakotondrainibe 3127, P00067229. Photos: G. Rouhan/MNHN.

Figure 4 Distribution map of Elaphoglossum andohahelense.

Red dots represent localities of specimens, and open circles represent the six main cities in Madagascar; the five altitudinal ranges corresponding globally to those generally recognized in Madagascar (Humbert, 1955; Faramalala, 1995) are represented in green (0–400 m), yellow (400–800 m), light brown (800–1,200 m), medium brown (1,200–1,800 m), and dark brown (>1,800 m).

Type:–MADAGASCAR. Toliara, Tolanaro (Fort-Dauphin), Eminiminy, parcelle 1, R.N.I. n 11 d’Andohahela versant Est et sommet du Trafon’omby: Camp 4, 15 km au NW du village d’Eminiminy, 24°34′15″S 46°43′85″E, 1,450–1,700 m, 18 Nov. 1995, F. Rakotondrainibe 3127 (holotype: P [P00067230!]; isotype: P [P00067229!]).

Diagnosis:–The rhizome scales, bright castaneous with subentire margins, makes Elaphoglossum andohahelense closer to E. ambrense Rouhan and E. anjanaharibense Rouhan (northern Madagascar), but distinctive characteristics of E. andohahelense include rhizome scales that are appressed on thinner rhizomes, laminar scales that are almost white, longer and lanceolate (vs. light red and stellate in E. anjanaharibense; castaneous to light red, and shorter to ovate in E. ambrense), and frond apices that are angustate (vs. cuneate to acute in E. ambrense).

Description:–Rhizomes long-creeping, 1.2–2.3 mm diam., branched, all over densely scaly (Figs. 3B and 3C); rhizome scales more or less antrorsely appressed, flat, acicular and broader at the base, 1.5–2.5 × 0.2–0.4 mm, castaneous, bright, translucent, brittle, peltate, bases round, apices long-attenuate, margins subentire sometimes with some short and irregular teeth (Fig. 3D).

Sterile fronds erect, inserted in two indistinct rows, 2–7 mm apart, 7–27 cm long (Fig. 3A); petioles 0.5–7.0 cm long, 0.7–1.0 mm diam., with scattered to overlapping scales; petiole scales deciduous, non-subulate, similar to the rhizome scales but pale, light red to white, matte, and papyraceous; sterile laminae herbaceous, linear, 7–21 × 0.4–1.3 cm, bases attenuate and long-decurrent in a narrow wing often to the petiole base, apices angustate, both surfaces with deciduous scales (persisting longer on abaxial surfaces) leaving glutinous dots when falling off; laminar scales scattered, non-subulate, lanceolate, 0.5–1.5 × 0.2–0.4 mm, light red to white, matte, papyraceous, peltate, bases round, apices acuminate to attenuate, margins subentire with often 2–4 short and irregular appendages at the base (Fig. 3E); median veins prominulous on both surfaces, rounded on abaxial surface, rounded or slightly sulcate and with angular ribs on adaxial surface, scales similar to those of laminae; lateral veins visible and slightly prominent on both surfaces, free, simple or bifurcate.

Fertile fronds unknown.

Etymology:–The specific epithet derives from the Malagasy name “Andohahela”, a mountain area rich in ferns, a national park, and the single locality known for the species; Andohahela is derived from “Andohan’ny-”, at the begnning of, and “-Tehela”, the name of a tribe from the village of Eminiminy located in the area of the Park (Pierrot Rabenandrasana & Lalao Andriamahefarivo, pers. com., 2020).

Habitat and distribution:–Endemic to Madagascar, E. andohahelense is known only from the type gathering and grows as epiphytic in wet evergreen forests from Andohahela area in the very South of Madagascar, in the Central phytogeographic domain, 1,450–1,700 m (Fig. 4).

Elaphoglossum anjanaharibense Rouhan, sp. nov. (Figs. 5 and 6)

Figure 5 Elaphoglossum anjanaharibense Rouhan.

(A) habit. (B) rhizome. (C) rhizome scale. (D) laminar scales. (E) detail of sterile lamina, abaxial surface. A–E, Rakotondrainibe 2547, P00046645. Photos: G. Rouhan/MNHN.

Figure 6 Distribution map of Elaphoglossum anjanaharibense.

Red dots represent localities of specimens, and open circles represent the six main cities in Madagascar; the five altitudinal ranges corresponding globally to those generally recognized in Madagascar (Humbert, 1955; Faramalala, 1995) are represented in green (0–400 m), yellow (400–800 m), light brown (800–1,200 m), medium brown (1,200–1,800 m), and dark brown (>1,800 m).

Type:–MADAGASCAR. Antsiranana, Andapa, Befingotra, RS d’Anjanaharibe-Sud, sur le versant Sud-Est, à 9.2 km à l’Ouest-Sud-Ouest de Befingotra, 1,300 m, 14°44′42″S 49°27′42″E, 28 Nov. 1994, F. Rakotondrainibe & F. Raharimalala 2547 (holotype: P [P00046645!]; isotypes: K, MO, TAN).

Diagnosis:–Elaphoglossum anjanaharibense is similar to other species with linear laminae and stellate scales on the fronds (i.e., E. lancifolium (Desv.) C.V.Morton, E. welwitschii (Baker) C.Chr., and E. aspidiolepis (Baker) C.Chr.): it differs by entire rhizome scales which are shorter and narrower than those of E. lancifolium, papyraceous to rigid and dark castaneous (vs. scarious and light brown in E. welwitschii from Africa and E. aspidiolepis from Madagascar), and the fertile fronds are shorter than the sterile ones (vs. fertile fronds longer in E. welwitschii and E. aspidiolepis). The rhizome scales are similar to those of E. ambrense Rouhan, but E. anjanaharibense differs by showing stellate, suborbicular scales on the fronds (vs. irregularly shaped, ovate to lanceolate, margins subentire to erose), narrower fronds <1 cm broad with angustate apices, fertile fronds shorter than the sterile ones, and it grows as epilithic in large populations on rocks in bed or on river banks.

Description:–Rhizomes long-creeping, 2–4 mm diam., glutinous, densely scaly (Fig. 5B); rhizome scales more or less patent, narrowly lanceolate to linear, 2.5–3.5 × 0.3–0.5 mm, dark castaneous, bright, translucent, papyraceous to rigid, bases truncate to cordate, apices attenuate, margins entire (Fig. 5C).

Sterile fronds erect, inserted in two distinct rows, 5–15 mm apart, 23–37 cm long (Fig. 5A); petioles 4–10 cm long, 0.9–1.3 mm diam., moderately to densely scaly, with the presence of red to black dots, rare to dense, non-glutinous, representing persistent points of attachment of scales; petiole scales non-subulate, appressed, 0.3–0.6 mm diam., stellate, with an orbicular to ovate scale body, light red, translucent, scarious, bases peltate or pseudopeltate, margins with cilia (usually >8) shorter than, or as long as the diameter of the scale body; sterile laminae herbaceous, linear, 19–30 × 0.6–1.0 cm, bases attenuate and decurrent, apices angustate, both surfaces glabrescent, with scales, and rare to dense, non-glutinous dots, reddish to black (Fig. 5E); laminar scales scattered, persistent on and along the median veins and on revolute margins, non-subulate, and similar to those of petioles (Fig. 5D); median veins prominent and rounded on both surfaces; lateral veins barely visible, free, simple or bifurcate, apices enlarged without hydathode.

Fertile fronds shorter than the sterile ones, laminae linear and with longer petioles (¼–⅓ of total frond length).

Etymology:–The specific epithet derives from the Malagasy name “Anjanaharibe-Sud” Special Reserve, a mountain area rich in ferns, and the single locality known for the species; Anjanaharibe is derived from “Zanahary”, God, and “be”, great, and the whole word means: where the great God is (Lalao Andriamahefarivo, pers. com., 2020).

Habitat and distribution:–Endemic to Madagascar, E. anjanaharibense is rare and grows as epilitic on river bank, in wet evergreen forests from Anjanaharibe-Sud (northern Madagascar) in the Central phytogeographic domain, 1,120–1,300 m (Fig. 6).

Additional specimens examined (paratypes):–MADAGASCAR. Rakotondrainibe 5020 (P).

Elaphoglossum approximatum Rouhan, sp. nov. (Figs. 7 and 8)

Figure 7 Elaphoglossum approximatum Rouhan.

(A) Habit. (B) Rhizome. (C) Rhizome scale. (D) Laminar scales, abaxial surface. (E) Detail of sterile lamina, abaxial surface. A–E, Antilahimena 4552, P06489142. Photos: G. Rouhan/MNHN.

Figure 8 Distribution map of Elaphoglossum approximatum.

Red dots represent localities of specimens, and open circles represent the six main cities in Madagascar; the five altitudinal ranges corresponding globally to those generally recognized in Madagascar (Humbert, 1955; Faramalala, 1995) are represented in green (0–400 m), yellow (400–800 m), light brown (800–1,200 m), medium brown (1,200–1,800 m), and dark brown (>1,800 m).

Type:–MADAGASCAR. Antsiranana, Vohemar, Daraina, Forêt de Binara, à 7.5 km au sud-ouest de Daraina, 13°15′12″S 49°37′12″E, 1,050 m, 4 Nov. 2001, F. Rakotondrainibe & H. Rasolohery 6486 (holotype: P [P00248588!]; isotype: TEF).

Diagnosis:–Elaphoglossum approximatum belongs to a group of four species (with E. rakotondrainibeae Rouhan, and E. longiacuminatum Rouhan, E. viridicaule Rouhan), having rhizomes with green apices in natura, and rhizome scales homogeneous, patent, to 5.5 mm long, dark castaneous with black hues, with entire margins, papyraceous and soft in texture. Elaphoglossum approximatum however differs from E. rakotondrainibeae and E. longiacuminatum by more closely spaced fronds, borne on shorter rhizomes. Unlike E. longiacuminatum, the laminae of E. approximatum are not long-acuminate at base, nor obtuse to round at apices; rhizomes are thicker than those of E. rakotondrainibeae; and the rhizome scales are long and soft, unlike those of E. viridicaule that are short, sclerotic and brittle.

Description:–Rhizomes short- to long-creeping, 3.2–4.9 mm diam., branched, showing green apices in natura, moderately to densely scaly (Fig. 7B); rhizome scales divergent and patent, narrowly lanceate to lanceolate, 2.5–5.5 × 0.4–0.8(–1.0) mm, dark castaneous with black hues, thicker and opaque towards the base and center, bright, translucent and papyraceous above (almost clathrate sometimes), peltate, bases round, apices attenuate to angustate, margins entire (Fig. 7C).

Sterile fronds erect, inserted in two distinct rows, 3–5(–7) mm apart (10–)15–34 cm long (Fig. 7A); petioles 3–14 cm long, 1–2 mm diam., glabrous or with some deciduous scales at the base; petiole scales non-subulate, similar to those of rhizomes; sterile laminae chartaceous to coriaceous, elliptic 7.0–21.0 × 1.5–4.0 cm, bases acuminate, apices acute to attenuate, both surfaces with inconspicuous scales (Fig. 7E); laminar scales scattered, non-subulate, reduced (<0.5 mm diam), irregularly arachnidoid, appressed, dark castaneous to black, quickly reduced to their point of attachment (Fig. 7D); median veins prominulous on both surfaces, rounded on the abaxial surface, slightly sulcate with rounded ribs on the adaxial surface; lateral veins barely or not visible, simple, 1–2-bifurcate, free, with apices reaching margins.

Fertile fronds about as long as the sterile ones, petioles as long or longer (3/10–½ of total frond length), and laminae narrower (<2 cm broad); sporangia not reaching margins, leaving all around a narrow, marginal strip.

Etymology:–The specific epithet derives from the Latin verb approximo, with its past participle approximatus meaning approximate. It is for the closely spaced fronds of the rhizomes, especially compared to E. rakotondrainibeae Rouhan, a species otherwise morphologically similar to E. approximatum.

Habitat and distribution:–Endemic to Madagascar, E. approximatum is rare and grows as epiphytic (rarely terrestrial) in wet evergreen forests from Northern Madagascar, in the Central phytogeographic domain, 780–1,546 m (Fig. 8).

Additional specimens examined (paratypes):–MADAGASCAR. Antilahimena 4552 (P), 4739 (P). - Rakotondrainibe 3414 (P), 6374bis (P). - Rakotovao 3558 (P). - Ravelonarivo 1627 (P).

Elaphoglossum brachymischum Rouhan, sp. nov. (Figs. 9 and 10)

Figure 9 Elaphoglossum brachymischum Rouhan.

(A) Habit. (B) Rhizome. (C) Rhizome scale. (D) Laminar scales. (E) Petiole. (F) Detail of sterile lamina, abaxial surface. A–F, Rakotondrainibe 2958, P00067053. Photos: G. Rouhan/MNHN.

Figure 10 Distribution map of Elaphoglossum brachymischum.

Red dots represent localities of specimens, and open circles represent the six main cities in Madagascar; the five altitudinal ranges corresponding globally to those generally recognized in Madagascar (Humbert, 1955; Faramalala, 1995) are represented in green (0–400 m), yellow (400–800 m), light brown (800–1,200 m), medium brown (1,200–1,800 m), and dark brown (>1,800 m).

Type:–MADAGASCAR. Toliara, Tolanaro (Fort-Dauphin), Eminiminy, parcelle 1, R.N.I. n 11 d’Andohahela versant Est et sommet du Trafon’omby: 8 km au NW du village d’Eminiminy, piste Camp1 vers Camp2, 24°37′55″S 46°45′92″E, 800 m, 28 Oct. 1995, F. Rakotondrainibe 2958 (holotype: P [P00067053!]).

Diagnosis:–Elaphoglossum brachymischum is similar to E. pseudovillosum Bonap. although the two species are known from two areas far apart from each other, and with distinct habitats: E. brachymischum was collected in a wet evergreen forest at 800 m asl in the very South of Madagascar, and E. pseudovillosum in a mountain sclerophyll forest at 1,500 m asl in the North.

Elaphoglossum brachymischum clearly differs by short-petiolate fronds (<⅕ of total frond length vs. > 2/5 in E. pseudovillosum), rhizome scales clearer, shorther and narrower (1.5–2.5 × 0.2–0.4 mm vs. 2.5–4.5 × 0.5–0.8 mm), laminae thinner and herbaceous (vs. chartaceous), and laminar scales with several long, ciliform, marginal appendages above the base (vs. margins subentire above the base, or with a few teeth).

Description:–Rhizomes short-creeping, 3.5–4.0 mm diam., unbranched, densely scaly especially towards apices (Fig. 9B); rhizome scales slightly patent, narrowly lanceate to linear, 1.5–2.5 × 0.2–0.4 mm, brown to dark castaneous, more or less bright, translucent, papyraceous, base truncate to cordate, apices cuneate to angustate, margins entire (Fig. 9C).

Sterile fronds erect to spreading, inserted in two distinct rows, <4 mm apart, 8–19 cm long (Fig. 9A); petioles 0.8–1.2 mm diam., 1–3 cm long (1/10–⅕ of total frond length), densely scaly but glabrescent (Fig. 9E); petiole scales more or less patent and divergent, non-subulate, similar to those of rhizomes but clear red, matte, scarious, peltate, margins with 2–5 teeth or short cilia (<0.5 mm long), apices long-attenuate to angustate; sterile laminae herbaceous, elliptic, 7–18 × 1.3–2.5 cm, bases cuneate sligthtly acuminate, apices acuminate, both surfaces and margins scaly but glabrescent (Fig. 9F); laminar scales scattered, imbricate on margins, non-subulate, lanceate to linear, 1–2 × 0.2–0.4 mm, red, translucent, scarious, bases peltate, apices long-attenuate to angustate, margins with 1–4(–5) ciliform appendages, straight and enlarged at their base, 0.5–0.8 mm long (i.e., longer than the scale body width) (Fig. 9D); median veins scaly, prominulous and rounded on the abaxial surface, flat to slightly sulcate (with rounded ribs) on the adaxial surface; median vein scales similar to those of laminae but larger, to 3 mm long; lateral veins visible, free, simple or bifurcate, apices enlarged without hydathodes.

Fertile fronds unknown.

Etymology:–The specific epithet combines the Greek prefix brachy-, short, with the name mischos, petiole; it refers to the short-petiolate fronds of the species.

Habitat and distribution:–Endemic to Madagascar, E. brachymischum is known only from the type gathering and grows as epiphytic in wet evergreen forest from the very South of Madagascar, in the Central phytogeographic domain, 800 m (Fig. 10).

Elaphoglossum cerussatum Tardieu subsp. brunneum Rouhan, subsp. nov. (Figs. 11 and 12)

Figure 11 Elaphoglossum cerussatum Tardieu subsp. brunneum Rouhan.

(A) Habit. (B) Rhizome. (C) Rhizome scales. (D) Laminar scales. (E) Petiole. (F) Detail of sterile lamina, adaxial surface. (G) Detail of sterile lamina, margin with continuous vein. A and B, F and G, Rakotondrainibe 2469, P00046938. C, Rakotondrainibe 3611, P00085177. D, Rakotondrainibe 3611, P00085177 (left), Rakotondrainibe 5189, P00181187 (right). E, Janssen 2874, P00590927. Photos: G. Rouhan/MNHN.

Figure 12 Distribution map of Elaphoglossum cerussatum Tardieu subsp. brunneum Rouhan.

Red dots represent localities of specimens, and open circles represent the six main cities in Madagascar; the five altitudinal ranges corresponding globally to those generally recognized in Madagascar (Humbert, 1955; Faramalala, 1995) are represented in green (0–400 m), yellow (400–800 m), light brown (800–1,200 m), medium brown (1,200–1,800 m), and dark brown (>1,800 m).

Type:–MADAGASCAR. Antsiranana, Andapa, RNI 12 du Marojejy, à 10.5 km au Nord-Ouest de Manantenina, sur un affluent de la rivière Andranomifototra, 14°26′24″S 49°44′30″E, 1520 m, 5 Nov. 1996, F. Rakotondrainibe 3611 (holotype: P [P00085177!]).

Diagnosis:–Elaphoglossum cerussatum subsp. brunneum differs from the type subspecies by the rhizome scales which are not white or light orange brown, but castaneous, matte (sometimes slightly bright), usually broader ((0.5)0.7–1.5 mm) (Fig. 11C), and arranged in layers such that scales are hardly discernable and separable from each other (Fig. 11B). Additionnaly, frond scales in E. cerussatum subsp. brunneum are light orange brown (vs. white in the type subspecies) and laminar scales are more regularly stellate (Figs. 11D and 11F).

Elaphoglossum cerussatum subsp. brunneum is morphologically close to the African species E. welwitschii (Baker) C.Chr., considering the stellate scales on laminae and the rhizome scales castaneous; it differs however from the latter by broader laminae >2 cm, with an intramarginal vein (Fig. 11G), and laminar apices often round (rarely obtuse to acute) (Fig. 11A).

Etymology:–The subspecific epithet derives from the Latin adjective brunneus, brown; it refers to the color of rhizome scales that are darker than those of the type subspecies.

Habitat and distribution:–Endemic to Madagascar, Elaphoglossum cerussatum subsp. brunneum is rare and grows as epiphytic in wet evergreen forests from the North (Tsaratanana, Marojejy and Anjanaharibe-Sud), in the Central phytogeographic domain, 1,520–2,063 m (Fig. 12). The two subspecies, cerussatum (type subspecies) and brunneum, co-occur in Marojejy and Anjanaharibe-Sud without growing truly in sympatry as they have different altitudinal distributions: subsp. cerussatum is locally frequent up to 1,400 m asl, and subsp. brunneum grows only from that altitude. The type gathering of the species E. cerussatum Tardieu, H. Humbert 22471, was collected around 1,400 m asl., and shows the altitudinal transition point between the two subspecies, as the two existing specimens at P (P00522341!) and K (K001208541!) represent E. cerussatum subsp. cerussatum and E. cerussatum subsp. brunneum, respectively.

Additional specimens examined (paratypes):–MADAGASCAR. Janssen 2874 (P). - Rakotondrainibe 2469 (P), 2480 (P), 3701 (P), 5166 (P), 5189 (P). - Rasolohery 376 (P), 377 (P).

Elaphoglossum coracinolepis Rouhan, sp. nov. (Figs. 13 and 14)

Figure 13 Elaphoglossum coracinolepis Rouhan.

(A) Habit. (B) Rhizome. (C) Rhizome scale. (D) Petiole scales. (E) Laminar scales. (F) Detail of sterile lamina, abaxial surface. (G) Habit in natura. A, D–E, Rasolohery 679, P00327983. B, F–G, Rouhan 353, P00749307. C, Rakotondrainibe 1954, P0061442. Photos: G. Rouhan/MNHN.

Figure 14 Distribution map of Elaphoglossum coracinolepis.

Red dots represent localities of specimens, and open circles represent the six main cities in Madagascar; the five altitudinal ranges corresponding globally to those generally recognized in Madagascar (Humbert, 1955; Faramalala, 1995) are represented in green (0–400 m), yellow (400–800 m), light brown (800–1,200 m), medium brown (1,200–1,800 m), and dark brown (>1,800 m).

Type:–MADAGASCAR. Masoala, Ambanizana, piste menant au sommet d’Ambohitsitondroina, 1,105 m, 15°34′31″ S 50°0′37″E, 23 Oct. 2004, G. Rouhan, T. Janssen, P. Antilahimena, D. Jean-Claude 353 (holotype: P [P00749307!]; isotypes: G, NBG, NY, P [P00749306!], TEF).

Diagnosis:–Elaphoglossum coracinolepis is similar to E. cerussatum Tardieu, considering the oblong laminae with round apices, and the continuous intramarginal veins of laminae, but Elaphoglossum coracinolepis unambiguously differs by the black scales scattered on laminae (vs. white or light orange brown in E. cerussatum).

Description:–Rhizomes short-creeping, 6–8 mm diam., unbranched, densely scaly (Fig. 13B); rhizome scales appressed and imbricate such that scales are hardly discernable and separable from each other, ovate to lanceolate, 1.5–2.5 × 0.5–1.0 mm, brown, matte, scarious, bases cordate to auriculate, apices acute to obtuse, margins somewhat lacerate with some irregular, non-glandular appendages (Figs. 13B and 13C).

Sterile fronds erect, inserted in two distinct rows, <3 mm apart (13–)20–55 cm long (Fig. 13A); petioles 5–13 cm long, 1.3–2.8 mm diam., dark brown, deciduously scaly; petiole scales dense to scattered, spreading to patent, non-subulate, narrowly lanceate to linear, 2–4 × 0.3–0.5 mm, concolorous (castaneous or black), or bicolorous with the lower part darker, bright, translucent (castaneous parts) to opaque (black parts), margins serrulate (Fig. 13D); sterile laminae chartaceous to coriaceous, oblong, 15–42 × 2.5–4.0 cm, bases cuneate to obtuse, apices rounded (rarely obtuse), both surfaces moderately to densely scaly but the adaxial surface and margins glabrescent (Fig. 13F); laminar scales of two kinds but all non-subulate (Fig. 13E): first ones scattered on laminae and denser along margins, lanceate to linear, 0.7–1.0(1.5) × 0.2–0.3 mm, light castaneous, translucent, and upper part sometimes blackish; second ones similar to the first ones, scattered on laminae and denser along median veins (absent from the margins), black but clearer around the point of attachment, bright, opaque, 0.7–1.5 × 0.2–0.5 mm; median veins prominulous on both surfaces, round on the abaxial surface, sulcate on the adaxial surface with scales similar to the second kind of the lamina; lateral veins visible, simple or 1–2-bifurcate, free but apices anastomosed into a continuous intramarginal vein.

Fertile fronds slightly shorter than, or as long as the sterile ones, long-petiolate (⅖–½ of total frond length); laminae linear, narrower than those of sterile ones, 1.1–1.7 cm broad.

Etymology:–The specific epithet derives from the Greek adjective coracinus, black raven, and name lepis, scale; it refers to the singular laminar scales that are black.

Habitat and distribution:–Endemic to Madagascar, E. coracinolepis is rare and grows as epiphytic in wet evergreen forests from the North and Center-North, in the Central phytogeographic domain, 850–1,450 m (Fig. 14).

Additional specimens examined (paratypes):–MADAGASCAR. Rakotondrainibe 1954 (P). - Rasolohery 679 (P).

Elaphoglossum desireanum Rouhan, sp. nov. (Figs. 15 and 16)

Figure 15 Elaphoglossum desireanum Rouhan.

(A) Habit. (B) Rhizome. (C) Rhizome scales. (D) Laminar scales. (E) Detail of sterile lamina with intramarginal vein, abaxial surface. (F) Habit in natura. A–E, Rakotondrainibe 6531, P00248637. F, Rouhan 378, P00749335. Photos: G. Rouhan/MNHN.

Figure 16 Distribution map of Elaphoglossum desireanum.

Red dots represent localities of specimens, and open circles represent the six main cities in Madagascar; the five altitudinal ranges corresponding globally to those generally recognized in Madagascar (Humbert, 1955; Faramalala, 1995) are represented in green (0–400 m), yellow (400–800 m), light brown (800–1,200 m), medium brown (1,200–1,800 m), and dark brown (>1,800 m).

Type:–MADAGASCAR. Antsiranana, Vohemar, Daraina, forêt de Binara à 7.5 km au sud-ouest de Daraina, 13°15′12″S 49°37′12″E, 1,000 m, 6 Nov. 2001, F. Rakotondrainibe & H. Rasolohery 6531 (holotype: P [P00248637!]; isotype: TEF).

Diagnosis:–Elaphoglossum desireanum can be easily confused with E. scolopendriforme Tardieu but it differs from the latter species by lateral veins more closely spaced, by a lower angle at which these lateral veins form an angle with the median vein, by an intramarginal vein discontinuous or absent, by thicker laminae, by rhizome scales larger, and by fronds usually longer-petiolate. Elaphoglossum desireanum more clearly differs from E. achroalepis (Baker) C.Chr by the absence of a continuous intramarginal vein, and by the rhizome scales that are light brown, ovate to lanceolate, subentire, broader and (vs. whitish to light red, narrowly lanceolate, 0,3–1,0 mm broad, margins with numerous, pluricellular, long appendages).

Description:–Rhizomes short-creeping, 4.0–7.5 mm diam., rarely branched, glossy black, glutinous, densely scaly (Fig. 15B); rhizome scales ovate to lanceolate, 4.0–6.5 × 1.0–1.8 mm, light brown, matte, scarious, peltate, bases round or cordate, apices acute to obtuse, margins subentire with sometimes a few irregular teeth (Fig. 15C).

Sterile fronds erect, inserted in two distinct rows, <5 mm apart, 15–41 cm long (Fig. 15A); petioles 2–14 cm long, 0.8–1.8 mm diam., deciduously scaly; petiole scales scattered, non-subulate, lanceolate, similar to those of rhizomes but narrower; sterile laminae herbaceous to chartaceous, narrowly lanceolate, oblong or elliptic 19.0–23.5 × 1.3–2.5(–3.5) cm, bases cuneate to acuminate, apices cuneate to attenuate, margins often undulate, both surfaces subglabrous (Fig. 15E); laminar scales non-subulate, minute, obscure and deciduous (quicker on the adaxial surface), scattered, light castaneous to white, 0.3–1.0 × 0.2–0.5 mm, lanceolate to substellate with some marginal, pluricellular, non-glandular appendages (Fig. 15D); median veins prominulous on both surfaces, rounded on the abaxial surface, slightly sulcate on the adaxial surface with same scales as those on the laminae; lateral veins visible, forming and angle at 40–50° with median veins, simple or 1–2-bifurcate, 1–2 mm apart (at base), free with sometimes a few anastomoses between adjacent veins, apices ending <0.2 mm from the laminar margin, enlarged without hydathodes, and usually free or sometimes anastomosed into a discontinuous intramarginal vein (e.g., Rakotondrainibe 6531) (Fig. 15E).

Fertile fronds slightly shorter than the sterile ones, longer-petiolate (⅓–½ of total frond length); fertile laminae linear, <1 cm broad; sporangia not reaching margins, leaving all around a narrow, marginal strip.

Eponymy:–The specific epithet honors our friend Désiré Ravelonarivo, Malagasy botanist working for MBG-Madagascar; he is native of the Andapa area where the species is well represented.

Habitat and distribution:–Endemic to Madagascar, E. desireanum is rare and grows as epiphytic in wet evergreen forests from the North, in the Eastern and Central phytogeographic domains, 620–1,600 m (Fig. 16).

Additional specimens examined (paratypes):–MADAGASCAR. Humbert 31757 (P). - Rakotondrainibe 2441 (P), 3654 (P), 5137bis (P), 5137 (P), 6485 (P). - Rouhan 378 (P), 379 (P).

Elaphoglossum glabricaule Rouhan, sp. nov. (Figs. 17 and 18)

Figure 17 Elaphoglossum glabricaule Rouhan.

(A) Habit. (B) Rhizome. (C) Detail of sterile lamina, abaxial surface. (D) Habit in natura. A, Antilahimena 7761, MO2980324. B–D, Antilahimena 7761, P06488838. Photos: G. Rouhan/MNHN (except D: P. Antilahimena/MBG-Madagascar).

Figure 18 Distribution map of Elaphoglossum glabricaule.

Red dots represent localities of specimens, and open circles represent the six main cities in Madagascar; the five altitudinal ranges corresponding globally to those generally recognized in Madagascar (Humbert, 1955; Faramalala, 1995) are represented in green (0–400 m), yellow (400–800 m), light brown (800–1,200 m), medium brown (1,200–1,800 m), and dark brown (>1,800 m).

Type:–MADAGASCAR. District Brickaville, commune Maroseranana, Fokontany Ambodilendemy, Ankerana, 1,179 m, 18°24′46″S 48°47′31″E, 23 Mar. 2011, P. Antilahimena, Félix, J. Randrimitantsoa 7761 (holotype: P [P06488838!]; isotype: MO [MO-2980324!]).

Diagnosis:–Elaphoglossum glabricaule is similar to E. humbertii C.Chr. and E. perrierianum C.Chr. in having thin, glabrous and glutinous rhizomes. But it differs from the two latter species by closer spaced fronds that are also shorther, narrower and subglabrous (vs. densely scaly in E. perrierianum, and with glutinous dots in E. humbertii); E. glabricaule differs also from E. humbertii by green rhizomes in natura (vs. black in E. humbertii).

Description:–Rhizomes long-creeping, 1.0–1.3 mm diam., glabrous, glossy, green irregularly smeared with black in natura (castaneous to black in herbarium), glutinous, almost glabrous, bearing only a few minute and inconspicuous squamules <0.2 mm (Fig. 17B); squamules irregularly shaped, round to ovate, dark red.

Sterile fronds erect, inserted in two distinct rows, 3–5 mm apart, up to 21 cm long, subsessile (petioles not distinct) (Fig. 17A); phyllopodia conspicuous, black and glutinous, 1–4 mm long; sterile laminae chartaceous, narrowly linear, 1.0–2.3 mm broad, bases long-attenuate and decurrent in a narrow wing to the phyllopodia, apices angustate, glabrous on both surfaces except some minute, irregular squamules similar to those on rhizomes (Fig. 17C); median veins prominulous on both surfaces, round on the abaxial surface, round to flat on the adaxial surface; lateral veins barely or not visible, simple, free, apices enlarged without hydathodes or anastomoses.

Fertile fronds slightly shorter than the sterile ones, ca. 10 cm long, long-petiolate (>⅗ of total frond length); laminae broader, up to 4 mm.

Etymology:–The specific epithet derives from the Latin adjective glaber, glabrous, and name caulis, stem; it refers to the glabrous rhizomes.

Habitat and distribution:–Endemic to Madagascar, E. glabricaule is known only from the type gathering (Ankerana), and grows as epiphytic in wet evergreen forest from the Center, in the Central phytogeographic domain, 1,072–1,179 m (Fig. 18).

Elaphoglossum gladiifolium Rouhan, sp. nov. (Figs. 19 and 20)

Figure 19 Elaphoglossum gladiifolium Rouhan.

(A) Habit. (B and C) Rhizome. (D) Rhizome scale. (E) Detail of sterile lamina, adaxial surface. (F) Detail of sterile lamina, abaxial surface. (G) Habit in natura. A and B, Rouhan 486, P00749466. C, F, Rouhan 1136, P02432650. D, Rouhan 296, P00749223. E, Rouhan 296, P00749228. G, Rouhan 1168, P02432690. Photos: G. Rouhan/MNHN.

Figure 20 Distribution map of Elaphoglossum gladiifolium.

Red dots represent localities of specimens, and open circles represent the six main cities in Madagascar; the five altitudinal ranges corresponding globally to those generally recognized in Madagascar (Humbert, 1955; Faramalala, 1995) are represented in green (0–400 m), yellow (400–800 m), light brown (800–1,200 m), medium brown (1,200–1,800 m), and dark brown (>1,800 m).

Type:–MADAGASCAR. Région d’Anosy, à proximité de Fort Dauphin, massif forestier surplombant le domaine St Jacques, forêt d’Ankarambilo [24°58′S 46°56′E], 550 m, 27 Nov. 2004, G. Rouhan, F. Rakotondrainibe, E. Randrianjohany 486 (holotype: P [P00749466!]; isotypes: G!, K!, NBG!, NY!, P [P00749465!], TEF).

Diagnosis:–Elaphoglossum gladiifolium Rouhan differs from other Malagasy species having also numerous glutinous dots on fronds, by having scaly rhizomes (vs. glabrous in E. humbertii C.Chr.), by laminae without scales (vs. scaly laminae in E. patriceanum Rouhan and E. lancifolium (Desv.) C.V.Morton), by fronds with attenuated apices and bases, and larger rhizome scales (vs. fronds with round to obtuse apices and attenuate bases, and smaller rhizome scales in E. subglabricaule Rouhan). In addition, E. gladiifolium differs from E. coriaceum Bonap. in growing as dense tufts, with herbaceous fronds with glutinous dots, and median veins prominulous on the adaxial side; laminae without scales, and rhizome scales dark, bright and brittle, prevent confusion with E. scolopendriforme Tardieu.

Description:–Rhizomes short-creeping, 3.0–4.5 mm diam., branched, dark castaneous to black, glutinous and glossy, densely scaly towards apices (Figs. 19B and 19C); rhizome scales lanceate to lanceolate, 1.5–3.5 × 0.5–0.8 mm, sclerotic, black or dark ferrugineous, bright, opaque or slightly translucent, bases round or cordate, apices cuneate, margins subentire or serrulate with short and irregular teeth (Fig. 19D).

Sterile fronds erect to spreading, inserted in two or three distinct rows, <7 mm apart (10–)20–60 cm long (Fig. 19A); petioles short and inconspicuous (fronds subsessile) to 15 cm long with laminae long-decurrent into a narrow wing often to the base, 1–2 mm diam., with numerous glutinous dots, ferrugineous and translucent turning sometimes black and opaque; some scales at the base only, similar to those of rhizomes; sterile laminae herbaceous, linear to narrowly elliptic, 10–50 × (0.5–)1.0–2.0 cm, bases long-acuminate to attenuate, apices attenuate, margins undulate, both surfaces without scales, with numerous glutinous dots at first ferrugineous, bright and translucent, turning sometimes whitish or blackish, matte and opaque (Figs. 19E and 19F); median veins prominulous on both surfaces, round on the abaxial surface (Fig. 19F), little or not sulcate (with angular ribs) on the adaxial surface (Fig. 19E); lateral veins visible, slightly prominulous on both surfaces, simple or bifurcate, free, apices submarginal enlarged without hydathodes or intramarginal vein.

Fertile fronds slightly shorter than the sterile ones, longer petiolate (⅖–⅗ of total frond length); laminae narrower.

Etymology:–The specific epithet derives from the Latin name gladius, sword, and folium, leaf; it refers to the frond shape.

Note:–Size, texture, and colors of the rhizome scales are variable among individuals (e.g., specimens of the gathering Rakotondrainibe 5887).

Habitat and distribution:–Endemic to Madagascar, E. gladiifolium is rare to frequent, and grows as epiphytic or epilithic in wet evergreen forests from North to South, in the Eastern and Central phytogeographic domains, 550–2,063 m (Fig. 20).

Additional specimens examined (paratypes):–MADAGASCAR. Bauret 143 (P), 157 (P). - Decary 14216 (P), 14693 (P), 14703 (P), 16778 (P), 18320 (P). - Humbert 17718 (P), 18184 (P), 22268 (P), 22448 (P), 22900 (P), 23074 (P), 28433 (P). - Janssen 2863 (P). - Malcomber 1764 (P). - Nusbaumer 2479 (P). - Perrier de la Bâthie 6137 (P), 7515 (P). - Rakotondrainibe 1495 (P), 1605 (P), 1945 (P), 1953bis (P), 2395 (P), 2579 (P), 3459 (P), 3581bis (P), 3581 (P), 3817 (P), 4077 (P), 4219 (P), 4298 (P), 4817 (P), 4846 (P), 5110bis (P), 5193 (P), 5887 (P), 6116 (P), 6137 (P), 6426 (P), 6492 (P). - Rasolohery 249 (P), 343 (P), 440 (P), 804 (P). - Razakamalala 3239 (P). - Razanajatovo 52 (P). - Rouhan 296 (P), 306 (P), 324 (P), 346 (P), 357 (P), 1136 (P), 1168 (P), 1274 (P).

Elaphoglossum leucolepis (Baker) Krajina ex Tardieu subsp. nanolepis Rouhan, subsp. nov. (Figs. 21 and 22)

Figure 21 Elaphoglossum leucolepis (Baker) Krajina ex Tardieu subsp. nanolepis Rouhan.

(A) Habit. (B and C) Rhizome. (D) Rhizome scale. (E) Petiole. (F) Petiole scale. (G) Detail of sterile lamina, abaxial surface. (H) Detail of sterile lamina, adaxial surface. (I) Laminar scales, abaxial surface. (J) Laminar scales, adaxial surface. A, D–J, Rouhan 322, P00749267. B and C, Rouhan 322, P00749266. Photos: G. Rouhan/MNHN.

Figure 22 Distribution map of Elaphoglossum leucolepis ssp. nanolepis.

Red dots represent localities of specimens, and open circles represent the six main cities in Madagascar; the five altitudinal ranges corresponding globally to those generally recognized in Madagascar (Humbert, 1955; Faramalala, 1995) are represented in green (0–400 m), yellow (400–800 m), light brown (800–1,200 m), medium brown (1,200–1,800 m), and dark brown (>1,800 m).

Type:–MADAGASCAR. Région de Diana, District d’Antsiranana II, Grand Lac, bords de forêt autour du lac, 12°35′48″S 49°9′34″ E, 1,300 m, 10 Oct. 2004, G. Rouhan, T. Janssen, H.L. Ranarijoana, E. Randrianjohany 322 (holotype: P [P00749267!]; isotypes: NY!, P [P00749266!], TEF!).

Diagnosis:–Elaphoglossum leucolepis subsp. nanolepis is similar to the type subspecies especially by the petiole scales which are bicolor with the same pattern (Figs. 21E and 21F), but E. leucolepis subsp. nanolepis differs by smaller laminar scales, 1–3 × 0.3–0.8 mm (vs. 3–5 × 0.8–1.5 mm), with <30 marginal cilia on each side (vs. 30–50), cilia 0.3–0.7 mm long (vs. <0.4 mm) representing 0.5–1.5 times the width of the scale body (vs. always shorter) (Figs. 21I and 21J). Additionaly, rhizomes (Figs. 21A and 21B), although also short-creeping, are lengthy and can be branched (vs. unbranched), and sterile fronds <40 cm long (vs. 20–60 cm) are sometimes less densely scaly.

Etymology:–The subspecific epithet derives from the Greek adjective nânos, small or smaller, and the name lepis, scale; it refers to the smaller size of the laminar scales compared to those of the type subspecies.

Habitat and distribution:–Endemic to Madagascar, Elaphoglossum leucolepis subsp. nanolepis is rare and grows as epiphytic in wet evergreen forests from the North, in the Central phytogeographic domain, 1,100–1,475 m (Fig. 22); this distribution area does not overlap with that of the other two subspecies, subsp. leucolepis and subsp. nigricans.

Additional specimens examined (paratypes):–MADAGASCAR. Decary 14663 (P). - Homolle 15 (P). - Humbert 32165 (P). - Malcomber 2340 (P). - McPherson 14492 (P). - Rakotondrainibe 1606 (P). - Rouhan 292 (P).

Elaphoglossum leucolepis (Baker) Krajina ex Tardieu subsp. nigricans Rouhan, subsp. nov. (Figs. 23 and 24)

Figure 23 Elaphoglossum leucolepis (Baker) Krajina ex Tardieu subsp. nigricans Rouhan.

(A) Habit. (B) Rhizome. (C) Rhizome scale. (D) Petiole. (E) Petiole scale. (F) Laminar scale. (G) Detail of sterile lamina, abaxial surface. (H) Detail of sterile lamina, adaxial surface. A, Rakotondrainibe 2990, P00067084. B–H, Rakotondrainibe 5808, P00181220. Photos: G. Rouhan/MNHN.

Figure 24 Distribution map of Elaphoglossum leucolepis ssp. nigricans.

Red dots represent localities of specimens, and open circles represent the six main cities in Madagascar; the five altitudinal ranges corresponding globally to those generally recognized in Madagascar (Humbert, 1955; Faramalala, 1995) are represented in green (0–400 m), yellow (400–800 m), light brown (800–1,200 m), medium brown (1,200–1,800 m), and dark brown (>1,800 m).

Type:–MADAGASCAR. Fianarantsoa, Ranomafana-Ifanadiana, PN de Ranomafana, forêt de Vatoharanana, à 4 km au sud-ouest du village de Ranomafana, 21°17′24″S 47°26′00″E, 980–990 m, 2 Oct. 2000, F. Rakotondrainibe, M. Randriambololona, H. Rasolohery, M. Rabarimanarivo, A. Rakotoarimanana 5808 (holotype: P [P00181220!]).

Diagnosis:–Elaphoglossum leucolepis, subsp. nigricans differs from the type subspecies by the petiole scales that are mostly concolorous, black to dark brown (only the marginal cilia are sometimes white) (Fig. 23E), by the presence of these scales also in the basal half (at least) of the abaxial median veins (Figs. 23A and 23G), and by often longer fronds, 34–80 cm (vs. 20–60 cm), with obtuse to acute apices.

Etymology:–The subspecific epithet derives from the Latin adjective nigricans, blackish, refering to the color of the petiole scales.

Habitat and distribution:–Endemic to Madagascar, Elaphoglossum leucolepis subsp. nigricans is rare and grows as pendent epiphytes in wet evergreen forests from Center-South and South (Ranomafana, Andringitra, Andohahela) in the Central phytogeographic domain, 800–1,400 m (Fig. 24). This distribution area does not overlap with that of the other two subspecies, leucolepis and nanolepis. Subspecies nigricans might have a preference for riverside habitats, as, at least 4 out of the 6 known gatherings were in this habitat.

Additional specimens examined (paratypes):–MADAGASCAR. Malcomber 2416 (P). - Rakotondrainibe 2600 (P), 2990 (P), 5830 (P), 6024 (P).

Elaphoglossum longiacuminatum Rouhan, sp. nov. (Figs. 25 and 26)

Figure 25 Elaphoglossum longiacuminatum Rouhan.

(A) Habit. (B) Rhizome. (C) Rhizome (arrows are for aerophores). (D) Rhizome scale. (E) Detail of sterile lamina, abaxial surface. (F) Laminar scale. A–C, F, Rouhan 1555, P02435817. D, Rouhan 1555, P02435818. E, Rouhan 1555, P02435014. Photos: G. Rouhan/MNHN.

Figure 26 Distribution map of Elaphoglossum longiacuminatum.

Red dots represent localities of specimens, and open circles represent the six main cities in Madagascar; the five altitudinal ranges corresponding globally to those generally recognized in Madagascar (Humbert, 1955; Faramalala, 1995) are represented in green (0–400 m), yellow (400–800 m), light brown (800–1,200 m), medium brown (1,200–1,800 m), and dark brown (>1,800 m).

Type:–MADAGASCAR. Parc national de Marojejy, Mont Beondroka, sommet, 14°26′15″S 49°48′14″E, 1,300 m, 20 Sep. 2015, G. Rouhan, L. Bauret, D. Ravelonarivo 1555 (holotype: P [P02435817!]; isotypes: P [P02435014!, P02435818!], TAN!).

Diagnosis:–Elaphoglossum longiacuminatum shows rhizome scales similar to those of E. approximatum Rouhan and E. rakotondrainibeae Rouhan (it is sympatric with the latter: see Rouhan et al., 1555 and Rouhan et al., 1557), but E. longiacuminatum differs by long-decurrent fronds, round to obtuse laminae at apices, median veins obviously ending before the laminar apices, and rhizomes thicker than those of E. rakotondrainibeae. The soft and longer rhizome scales make the distinction easier with E. viridicaule Rouhan. Elaphoglossum longiacuminatum share with E. rakotondrainibeae, E. sabineanum Rouhan, and E. ovalilimbatum Bonap., the long-creeping rhizomes with erect and widely spaced fronds, but it clearly differs from the two latter species by larger fronds and thicker rhizomes with erect and much longer rhizome scales. Outside Madagascar, the laminar shape and dark rhizome scales may remind the African E. angustatum (Schrader) Hieron., but the rhizome scales in E. longiacuminatum are entire, and darker, almost black.

Description:–Rhizomes long-creeping, 3.2–5.0 mm diam., branched, scaly (Fig. 25B), showing green apices in natura; rhizome scales scattered (denser towards apices), divergent and patent, narrowly lanceate to lanceolate (2.5–)3.0–5.0 × 0.5–1.0 mm, dark castaneous with black hues, thicker and opaques towards base and center, bright and translucent above (almost clathrate sometimes), peltate, bases round, apices attenuate to angustate, margins entire (Fig. 25D); aerophores present in pairs at the frond base (on phyllopodia or rhizomes), finger-shaped, green (Fig. 25C).

Sterile fronds erect, inserted in two distinct rows, 6–20 mm apart, 22–35 cm long (Fig. 25A); petioles 6–13 cm long, 1.2–3.1 mm diam., light green, with some deciduous scales in the basal half; petiole scales black, non-subulate, similar to those of rhizomes but smaller and with rare, short, marginal appendages towards the base; sterile laminae chartaceous to coriaceous, obovate, oblong or elliptic, 13–28 × 3.5–6.0 cm, bases long-acuminate, decurrent in a narrow wing to the petiole base, apices round or obtuse, both surfaces with inconspicuous scales (Fig. 25E); laminar scales scattered, non-subulate, appressed, reduced (<0.5 mm diam), irregularly arachnidoid, dark castaneous to black, quickly reduced to their point of attachment (Fig. 25F); median veins obviously ending before the laminar apices (0.5–1.0 cm from margins), prominulous on both surfaces, round on the abaxial surface, sulcate with rounded ribs on the adaxial surface; lateral veins little or not distinct, simple or 1–2-bifurcate, free to the marginal apices.

Fertile fronds longer than the sterile ones, to 70 cm long, longer-petiolate (⅗ of total frond length) and narrower, 2.5 cm broad; sporangia not reaching margins, leaving all around a narrow, marginal strip.

Etymology:–The specific epithet derives from the Latin adjectives longus, long, and acuminatus, acuminate; it refers to the shape of the base of sterile laminae.

Habitat and distribution:–Endemic to Madagascar, E. longiacuminatum is known only from the type gathering, and grows as terrestrial in wet evergreen, summit forest, from the North, in the Central phytogeographic domain, at 1,300 m (Fig. 26).

Elaphoglossum patriceanum Rouhan, sp. nov. (Figs. 27 and 28)

Figure 27 Elaphoglossum patriceanum Rouhan.

(A) Habit. (B) Rhizome. (C) Rhizome scales. (D) Petiole, abaxial surface. (E and F) Detail of sterile lamina, abaxial surface. (G and H) Detail of sterile lamina, adaxial surface. (I) Laminar scales. (J and K), Habit in natura. A–E, K, Rouhan 417, P00749380. F, Rouhan 1292, P02432856. G, I, Rouhan 1228, P02432772. H, Rouhan 410, P00749373. J, Rouhan 376, P00749333. Photos: G. Rouhan/MNHN.

Figure 28 Distribution map of Elaphoglossum patriceanum.

Red dots represent localities of specimens, and open circles represent the six main cities in Madagascar; the five altitudinal ranges corresponding globally to those generally recognized in Madagascar (Humbert, 1955; Faramalala, 1995) are represented in green (0–400 m), yellow (400–800 m), light brown (800–1,200 m), medium brown (1,200–1,800 m), and dark brown (>1,800 m).

Type:–MADAGASCAR. Région de Alaotra-Mangoro, District de Moramanga, Andasibe, Parc national de Mantadia, piste commençant au PK14 et s’élevant vers la crête, 13°50′44″S 48°26′17″E, 1,050 m, 12 Nov. 2004, G. Rouhan & T. Janssen 417 (holotype: P [P00749380!]; isotypes: K!, NY!, TEF).

Diagnosis:–Elaphoglossum patriceanum differs from E. coriaceum by the presence of laminar scales, numerous glutinous dots on fronds, by median veins sulcate but not immersed, laminae overall flat and not revolute, tip of laminar apices obtuse (vs. attenuate to angustate), and fertile fronds usually longer than the sterile ones (vs. shorter or equaling). Also similar to E. subglabricaule Rouhan, E. humbertii C.Chr. and E. gladiifolium Rouhan, E. patriceanum is distinguished by the presence of laminar scales, and rhizomes all over densely scaly. The substellate scales and glutinous dots on fronds might be compared to those of E. lancifolium (Desv.) C.V.Morton, but E. patriceanum shows smaller and darker rhizome scales, and coriaceous laminae with obtuse apices.

Description:–Rhizomes short-creeping, 2.5–5.0 mm diam., rarely branched, glossy, black, glutinous, all over densely scaly (Fig. 27B); rhizome scales imbricate, ovate to lanceate, 1.0–1.5 × 0.3–0.5 mm, sclerotic, black or dark ferrugineous, bright and opaque, bases round or cordate, apices acute, margins subentire or serrulate with basiscopic teeth (Fig. 27C).

Sterile fronds erect, inserted in two distinct rows, <5 mm apart, 5–20(–27) cm long (Fig. 27A); petioles short and inconspicuous (subsessile fronds) to 10 cm long with long-decurrent laminae, 0.9–1.5 mm diam., with scales and numerous glutinous dots ferrugineous to black (Fig. 27D); petiole scales scattered, of two kinds but all non-subulate: the first ones similar to those of rhizomes but shorter, ovate to short-oblong, 0.3–0.6 × 0.2–0.3 mm, margins subentire, and the second ones similar to those of laminae (see below); sterile laminae coriaceous and rigid, linear or narrowly oblanceolate, 5–17 × 0.5–1.0 cm, bases attenuate and decurrent, apices cuneate but obtuse to round at the very tip, both surfaces deciduously scaly (scales persisting longer on margins) and with numerous glutinous dots ferrugineous, at first bright and translucent turning sometimes whitish or black, matte and opaque (Figs. 27E–27H); laminar scales scattered, non-subulate, lanceate, margins with up to 6 acicular cilia on each side, 0.3–1.0 × 0.5–1.0 mm (cilia included, 1–4 times as long as the scale body), light castaneous, translucent, peltate (Fig. 27I); median veins prominulous on both surfaces, round on the abaxial surface, sulcate with rounded ribs on the adaxial surface, with scales scattered in the basal half, scales ovate to short-oblong, 0.3–0.6 × 0.2–0.3 mm, margins subentire; lateral veins little or not visible, simple or bifurcate, free, with apices reaching margins.

Fertile fronds as long as or slightly longer than the steriles ones, longer-petiolate (½–⅗ of total frond length); fertile laminae narrowly oblong, narrower than or as broad as the sterile ones, bases acute.

Eponymy:–The specific epithet honors our friend Patrice Antilahimena, Malagasy botanist working for MBG-Madagascar, collector of a paratype in the Ambatovy area where he has been much contributing to the knowledge of the flora.

Note:–Elaphoglossum patriceanum is sympatric –and thus has been confused– with E. humbertii C.Chr. and E. gladiifolium Rouhan, as illustrated by for example, Humbert 22448bis representing E. humbertii, and Humbert 22448 being a mixed gathering with specimens representing E. patriceanum and others E. gladiifolium.

Habitat and distribution:–Endemic to Madagascar, E. patriceanum is rare to locally frequent and grows as epiphytic in wet evergreen forests from the Nord and Center, in the Eastern and Central phytogeographic domains, 400–1,700 m (Fig. 28).

Additional specimens examined (paratypes):–MADAGASCAR. Antilahimena 3300A (P). - Cours 2318 (P), 2318 (P), 3346bis (P), 3346ter (P). - Deroin 37 (P). - Humbert 22448 (P), 22491 (P), 22541 (P), 23872 (P). – Jard. Bot. 2820 (P). - Nusbaumer 1643 (P). - Perrier de la Bâthie 6127 (P), 18270 (P). - Proisy 241 (P). - Rabarimanarivo 182 (P), 196 (P). - Rakotondrainibe 1966 (P), 1967 (P), 2130 (P), 2368 (P), 2369 (P), 2380 (P), 2389 (P), 3408 (P), 3485 (P), 3488 (P), 3489 (P), 3630bis (P), 3646 (P), 5073 (P), 5125 (P), 5973 (P), 6169 (P), 6327 (P), 6493 (P), 6533 (P), 6540 (P). - Rasolohery 262 (P), 552 (P). - Rouhan 361 (P), 376 (P), 410 (P), 1151 (P), 1228 (P), 1292 (P), 1504 (P), 1569 (P).

Elaphoglossum perangustum Rouhan, sp. nov. (Figs. 29 and 30)

Figure 29 Elaphoglossum perangustum Rouhan.

(A) Habit. (B) Rhizome. (C) Rhizome scale. (D) Detail of sterile lamina, abaxial surface. (E) Detail of sterile lamina, adaxial surface. (F) Laminar scales. (G) Habit in natura. A–F, Rouhan 1513, P02435813. G, Rouhan 1513, P02434961. Photos: G. Rouhan/MNHN.

Figure 30 Distribution map of Elaphoglossum perangustum.

Red dots represent localities of specimens, and open circles represent the six main cities in Madagascar; the five altitudinal ranges corresponding globally to those generally recognized in Madagascar (Humbert, 1955; Faramalala, 1995) are represented in green (0–400 m), yellow (400–800 m), light brown (800–1,200 m), medium brown (1,200–1,800 m), and dark brown (>1,800 m).

Type:–MADAGASCAR. Toamasina, Moramanga, site minier d’Ambatovy, Analamay, zone de conservation 23, au départ de cette piste descente vers le sud pour rejoindre la rivière, 18°48′28″S 48°20′07″E, 1,110 m, 12 Sep. 2015, G. Rouhan, L. Bauret, P. Antilahimena 1513 (holotype: P [P02435813!]; isotypes: P [P02434961!], TAN!).

Diagnosis:–Elaphoglossum perangustum resembles E. rakotondrainibeae Rouhan, E. longiacuminatum Rouhan, and E. approximatum Rouhan especially by the dark rhizome scales, but it differs by long-creeping rhizomes with rhizome scales scattered and dark brown to black, by widely spaced fronds, and narrow, linear laminae. Elaphoglossum perangustum differs from E. coursii Tardieu by longer, papyraceous and fully translucent rhizome scales, and by prominulous median veins that are not adaxially immersed.

Description:–Rhizomes long-creeping, 1.6–2.8 mm diam., branched, scaly, pale green in natura; rhizomes scales scattered (denser towards apices) (Fig. 29B), divergent and patent, narrowly lanceate to lanceolate, 3–5 × 0.3–0.6(–1.0) mm, mostly brown to black (color is sometimes heterogeneous among scales, and some scales are even light brown and dark brown), papyraceous, bright and translucent (almost clathrate sometimes), bases cordate to round, peltate, apices attenuate to angustate, margins entire or with a few ciliform and glandular appendages shorter than the scale width (Fig. 29C).

Sterile fronds erect, inserted in two indistinct rows (7–)10–20 mm apart, 17–40 cm long (Fig. 29A); petioles (1–)5–16 cm long, 0.8–1.3 mm diam., subglabrous with a few scales similar to those of rhizomes but smaller; sterile laminae chartaceous to coriaceous, linear, 13–29 × 0.4–1.0(–1.7) cm, bases attenuate and decurrent, apices angustate, both surfaces with inconspicuous scales (Figs. 29D and 29E); laminar scales scattered, non-subulate, reduced (<0.5 mm diam), irregularly arachnidoid, appressed, dark castaneous (Fig. 29F); median veins prominulous on both surfaces, round on the abaxial surface (Fig. 29D), sulcate with rounded ribs on adaxial surfaces (Fig. 29E); lateral veins indistinct, simple ou bifurcate, free.

Fertile fronds as long as the sterile ones, longer-petiolate (ca. 7/10 of total frond length).

Etymology:–The specific epithet derives from the Latin adjective perangustus, very narrow; it refers to the laminar width.

Habitat and distribution:–Endemic to Madagascar, E. perangustum is rare and grows as terrestrial in large populations (including some dozens of fronds), in evergreen forests from the Centre, in the Central phytogeographic domain, 950–1,450 m (Fig. 30).

Although being known by only four gatherings, E. perangustum might be more widely distributed because the species could have been mostly ignored in being confused with E. lepervanchei (Bory ex Fée) T.Moore which is one of the most frequent and very variable species including rather narrow fronds (rhizome scales are however distinct).

Additional specimens examined (paratypes):–MADAGASCAR. Rakotondrainibe 3727 (P), 5972 (P). - Rouhan 411 (P).

Elaphoglossum prominentinervulum Rouhan, sp. nov. (Figs. 31 and 32)

Figure 31 Elaphoglossum prominentinervulum Rouhan.

(A) Habit. (B) Rhizome. (C) Rhizome (arrows are for aerophores). (D and E) Rhizome scales. (F) Detail of sterile lamina with intramarginal vein, abaxial surface. (G) Detail of sterile lamina, adaxial surface. (H and I) Habit in natura. (J) Laminar scales. A, Rouhan 1192, P02432716. B, Rouhan 1582, P02435070. C, H, Rouhan 1192, P02432715. D, Rakotondrainibe 2455, P00046927. E, Rakotondrainibe 2776, P00075151. F and G, J, Rouhan 354, P00749308. I, Rouhan 354, P00749309. Photos: G. Rouhan/MNHN.

Figure 32 Distribution map of Elaphoglossum prominentinervulum.

Red dots represent localities of specimens, and open circles represent the six main cities in Madagascar; the five altitudinal ranges corresponding globally to those generally recognized in Madagascar (Humbert, 1955; Faramalala, 1995) are represented in green (0–400 m), yellow (400–800 m), light brown (800–1,200 m), medium brown (1,200–1,800 m), and dark brown (>1,800 m).

Type:–MADAGASCAR. Région de Sava, Maroantsetra, péninsule de Masoala, Parc national, piste menant du camps intermédiaire au sommet d’Ambanizana, Ambohitsitondroina, versant ouest, 15°34′31″S 50°0′37″E, 1,050 m, 23 Dec. 2004, G. Rouhan, T. Janssen, P. Antilahimena, D. Jean-Claude 354 (holotype: P [P00749309!]; isotypes: NY!, P [P00749308!], TEF).

Diagnosis:–Elaphoglossum prominentinervulum has been confused with E. angustatum (Schrad.) Hieron. (Africa), E. conforme (Sw.) J.Sm. (Africa and Madagascar), and E. acrostichoides (Hook. & Grev.) Schelpe (Africa and Western Indian Ocean), but it is distinguished by different rhizome scales, and especially by particularly prominulous lateral veins, tangible and visible on both sides, which look like Chinese shadow. This latter character is shared with E. sinensiumbrarum Rouhan sp. nov., but E. prominentinervulum is clearly distinguished by the rhizome scales, which are ovate to lanceolate, dark castaneous and sometimes partially sclerotic (vs. narrowly lanceolate to linear, light brown to stramineous, translucent).

Description:–Rhizomes long-creeping, 2.5–5.0 mm diam., rarely branched, moderately to densely scaly (Fig. 31B); rhizome scales closely to loosely appressed, ovate to lanceolate, 1.5–3.0(–4.0) × 0.5–1.5(–2.0) mm, papyraceous sometimes partially sclerotic, dark castaneous with sometimes darker spots (blackish, diffuse and irregularly distributed), bright, peltate, bases cordate-imbricate, apices acute to attenuate, margins with irregular multicellular appendages, which are ciliform but contorted, sometimes glandular, shorter than to slightly longer than the greater width of the body scale (Figs. 31D and 31E); presence of a pair of light green, finger-shaped aerophores at the base of each frond (Fig. 31C).

Sterile fronds erect, inserted in two distinct rows, 5–15 mm apart, 13–45(–55) cm long (Fig. 31A); petioles 3–18(–24) cm long, 1–2 mm diam., glabrous with only a few persisting scales at base and similar to those of rhizomes; sterile laminae chartaceous to coriaceous, elliptic, 10–31 × 1.7–4.5(–6.5) cm, bases acute to cuneate, apices acute to acuminate, margins thinner, clear and cartilaginous, both surfaces with inconspicuous scales which are deciduous on adaxial surfaces (Figs. 31F and 31G); laminar scales scattered, non-subulate, reduced, 0.1–0.3 mm diam, irregularly arachnidoid, appressed, dark castaneous to black (Fig. 31J); median veins prominulous on both surfaces, round and often bisulcate on abaxial surfaces (Fig. 31F), sulcate with rounded ribs on adaxial surfaces (Fig. 31G): lateral veins particularly distinct, prominulous, and tangible on both sides (Fig. 31I), and at 55–75° angle to median veins, simple or 1–2-bifurcate, 1.0–2.5 mm apart (at base), free but apices anastomosed into a continuous or discontinuous intramarginal vein.

Fertile fronds slightly shorter than or about as long as the sterile ones, elliptic to lanceolate, longer petiolate (⅖–⅘ of total frond length), somewhat narrower; sporangia not reaching margins, leaving all around a narrow, marginal strip.

Etymology:–The specific epithet derives from the Latin adjective prominens, prominulous, and name nervulus, vein; it refers to the lateral veins wich are particularly prominulous.

Habitat and distribution:–Endemic to Madagascar, E. prominentinervulum is little frequent and grows as epiphytic in evergreen forests from the North and Center, in the Central phytogeographic domain, 1,000–2,137 m d’altitude (Fig. 32).

Additional specimens examined (paratypes):–MADAGASCAR. Borie 512 (P). - Cours 3565 (P). - Humbert 17676 (P), 17894p.p. (P), 22576 (P), 22611bis (P), 23807 (P), 31401 (P), 31564 (P). - Rakotondrainibe 1962 (P), 2272 (P), 2289bis (P), 2361 (P), 2364 (P), 2426 (P), 2455 (P), 2481 (P), 2496 (P), 2737 (P), 3496 (P), 3514 (P), 3520 (P), 3550 (P), 3616bis (P), 3616 (P), 3664 (P), 4273 (P), 4980 (P), 4982 (P), 5132 (P), 5140 (P), 5182 (P), 6181 (P). - Rasolohery 685 (P), 861 (P), 865 (P). - Rouhan 303 (P), 320 (P), 323 (P), 1192 (P), 1582 (P).

Elaphoglossum rakotondrainibeae Rouhan, sp. nov. (Figs. 33 and 34)

Figure 33 Elaphoglossum rakotondrainibeae Rouhan.

(A) Habit. (B and C) Rhizome. (D) Rhizome scale. (E) Detail of sterile lamina, abaxial surface. (F) Laminar scales. (G) Habit in natura. A, D, F and G, Rouhan 1214, P02432749. B and C, E, Rouhan 1214, P02432750. Photos: G. Rouhan/MNHN.

Figure 34 Distribution map of Elaphoglossum rakotondrainibeae.

Red dots represent localities of specimens, and open circles represent the six main cities in Madagascar; the five altitudinal ranges corresponding globally to those generally recognized in Madagascar (Humbert, 1955; Faramalala, 1995) are represented in green (0–400 m), yellow (400–800 m), light brown (800–1,200 m), medium brown (1,200–1,800 m), and dark brown (>1,800 m).

Type:–MADAGASCAR. RNI 12 du Marojejy, à 11 km au Nord-Ouest de Manantenina, 14°26′12″S 49°44′30″E, 1,300 m, 30 Oct. 1996, F. Rakotondrainibe 3593 (holotype: P [P00085152!]; isotypes: P [P00085153!]; TEF).

Diagnosis:–Elaphoglossum rakotondrainibeae is distinguished from E. approximatum Rouhan by more widely spaced fronds, and by longer and thinner rhizomes; it is distinguished from the sympatric E. longiacuminatum Rouhan by thinner rhizomes, laminae short-acuminate non-decurrent at base and attenuate at apex (vs. laminae long-acuminate, decurrent in a narrow wing to the petiole base, and round or obtuse at apex), and median veins ending at the laminar apices (vs. ending before the laminar apices). It differs from E. viridicaule Rouhan by longer and soft rhizome scales. It is undoubtedly distinguished from E. sabineanum Rouhan and E. ovalilimbatum Bonap., which have much smaller rhizome scales, and round to acute laminar apices. Outside Madagascar, E. rakotondrainibeae differs from the African E. angustatum (Schrad.) Hieron. by darker, entire rhizome scales, and broader elliptic laminae with non decurrent bases and attenuate apices.

Description:–Rhizomes long-creeping, 1.2–2.5(–3.5) mm diam., branched, showing green apices in natura (Fig. 33C), with scales grouped in small clusters and thus irregularly scattered (Fig. 33B); rhizome scales divergent and patent, narrowly lanceate to lanceolate, 3.0–5.5 × 0.5–1.3 mm, dark castaneous with black hues, thicker and opaque towards base and center, bright, translucent and papyraceous above (almost clathrate sometimes), peltate, bases round, apices attenuate to angustate, margins entire or subentire (rare irregular appendages) (Fig. 33D); aerophores present at the frond base, finger-shaped, green (Fig. 33C).

Sterile fronds erect, inserted in two indistinct rows (5–)14–30(–60) mm apart, 6–32 cm long (Fig. 33A); petioles 2.5–11.0 cm long, 0.7–1.5 mm diam., light green, glabrous or with a few deciduous scales similar to those of rhizomes but narrower (0.2-0.3 mm broad), lanceate to linear; sterile laminae chartaceous to coriaceous, elliptic or slightly ovate, 3.5–18.0(–23.0) × 2–5(–6) cm, bases acuminate, apices attenuate (often abruptly), margins thinner, clear and cartilaginous, both surfaces with inconspicuous scales which are deciduous on adaxial surfaces (Fig. 33E); laminar scales scattered, non-subulate, irregularly arachnidoid, appressed, <0.5 mm diam, dark castaneous to black, soon reduced to their point of attachment (Figs. 33E and 33F); a few other laminar scales towards the base of median veins, linear, ca. 1.5 mm long; median veins prominulous on both surfaces, round on the abaxial surfaces, slightly sulcate with rounded ribs on adaxial surfaces (Fig. 33E); lateral veins slightly distinct, simple or 1–2-bifurcate, free.

Fertile fronds about as long as the sterile ones, longer petiolate (>½ of total frond length), laminae narrower (<0.9–1.7 cm); sporangia not reaching margins, leaving all around a narrow, marginal strip.

Eponymy:–The specific epithet honors our colleague, Dr. France Rakotondrainibe, who much contributed to the knowledge of pteridophytes in Madagascar, and especially in the Marojejy national Park (Rakotondrainibe, 2000; Rakotondrainibe et al., 2003) where she collected the type gathering.

Habitat and distribution:–Endemic to Madagascar, E. rakotondrainibeae is rare and grows as epiphytic in evergreen forests from the North, in the Central phytogeographic domain, 1,200–1,700 m (Fig. 34).

Additional specimens examined (paratypes):–MADAGASCAR. Humbert 31871 (P). - Rakotondrainibe 3468 (P), 4952 (P), 5000 (P). - Rouhan 1214 (P), 1557 (P).

Elaphoglossum repandum Rouhan, sp. nov. (Figs. 35 and 36)

Figure 35 Elaphoglossum repandum Rouhan.

(A) Habit. (B) Rhizome. (C) Rhizome scales. (D) Petiole. (E) Detail of sterile lamina, abaxial surface. (F) Detail of sterile lamina, adaxial surface. (G) Laminar scales. A–G, Rakotondrainibe 2918, P00067015. Photos: G. Rouhan/MNHN.

Figure 36 Distribution map of Elaphoglossum repandum.

Red dots represent localities of specimens, and open circles represent the six main cities in Madagascar; the five altitudinal ranges corresponding globally to those generally recognized in Madagascar (Humbert, 1955; Faramalala, 1995) are represented in green (0–400 m), yellow (400–800 m), light brown (800–1,200 m), medium brown (1,200–1,800 m), and dark brown (>1,800 m).

Type:–MADAGASCAR. Toliara, Tolanaro (Fort-Dauphin), Eminiminy, parcelle 1, R.N.I. n 11 d’Andohahela versant Est et sommet du Trafon’omby: 8 km au NW du village d’Eminiminy, 24°37′55″S 46°45′92″E, 500 m, 24 Oct. 1995, F. Rakotondrainibe 2918 (holotype: P [P00067015!]; isotype: TEF).

Description:–Rhizomes long-creeping, 0.9–1.7 mm diam., branched, all over densely scaly (Fig. 35B); rhizome scales more or less patent, lanceate, 0.8–1.5 × 0.2–0.4 mm, dark brown to castaneous, little bright, translucent with thicker and darker cell walls (almost clathrate), papyraceous to chartaceous, peltate, bases round, apices cuneate, margins entire (Fig. 35C).

Sterile fronds erect to spreading, inserted in two little distinct rows, 2–8 mm apart, 7–15 cm long (Fig. 35A); petioles 3–8 cm long, 0.5–0.8 mm diam., moderately to densely scaly (but scales not imbricate) (Fig. 35D); petiole scales patent, deciduous, non-subulate, narrowly lanceate to linear, 0.7–1.0 × 0.1–0.2 mm, red, matte or little bright, translucent, payraceous, peltate, bases round, apices angustate, margins entire; sterile laminae herbaceous, narrowly oblanceolate to oblong, 4.5–8.0 × 0.5–1.0 cm, bases attenuate and decurrent in a narrow wing, apices obtuse to round, margins irregulary wavy, both surfaces and margins deciduously scaly (Figs. 35E and 35F); laminar scales more or less patent, scattered, non-subulate, similar to those of petioles, to 1.5 mm long (Fig. 35G); median veins prominulous on both surfaces, round on the abaxial surfaces, round or slightly sulcate on the adaxial surfaces, with laminar scales; lateral veins little visible, simple or bifurcate, free, apices submarginal, enlarged and darker without hydathode.

Fertile fronds unknown.

Etymology:–The specific epithet derives from the Latin adjective repandus, irregularly wavy, and refers to the laminar margins.

Note:–Elaphoglossum repandum is easily recognized among other Malagasy species of the genus, by small, linear, red, and patent (not subulate) frond scales, and by fronds small, narrowly oblanceolate to oblong, with repand margins.

Habitat and distribution:–Endemic to Madagascar, E. repandum is known only from the type gathering and grows as epilithic in wet evergreen forests from Andohahela area in the very South of Madagascar, in the Central phytogeographic domain, 500 m (Fig. 36).

Elaphoglossum sabineanum Rouhan, sp. nov. (Figs. 37 and 38)

Figure 37 Elaphoglossum sabineanum Rouhan.

(A) Habit. (B and C) Rhizome. (D) Rhizome scales. (E) Detail of sterile lamina, abaxial surface. (F) Detail of sterile lamina, adaxial surface. (G) Laminar scales. A–G, Rakotondrainibe 5896, P00212379. Photos: G. Rouhan/MNHN.

Figure 38 Distribution map of Elaphoglossum sabineanum.

Red dots represent localities of specimens, and open circles represent the six main cities in Madagascar; the five altitudinal ranges corresponding globally to those generally recognized in Madagascar (Humbert, 1955; Faramalala, 1995) are represented in green (0–400 m), yellow (400–800 m), light brown (800–1,200 m), medium brown (1,200–1,800 m), and dark brown (>1,800 m).

Type:–MADAGASCAR. PN Ranomafana, forêt de Vatoharanana, 21°17′24″S 47°26′00″E, 1,050–1,060 m, 5 Oct. 2000, F. Rakotondrainibe, M. Randriambololona, H. Rasolohery, M. Rabarimanarivo, A. Rakotoarimanana 5896 (holotype: P [P00212379!]).

Diagnosis:–Elaphoglossum sabineanum differs from E. ovalilimbatum Bonap. by its fronds lanceolate to narrowly elliptic, most often long-decurrent in a narrow wing (vs. shortly oblong or elliptic, base obtuse short-acuminate at base), by its median veins prominulous on the adaxial surface, and by its wider distribution from North to Center (vs. North). Elaphoglossum sabineanum is more easily distinguished from E. rakotondrainibeae Rouhan by much smaller (0.5–1.2 × 0.2–0.6 mm), appressed and black rhizome scales (vs. 3.0–5.5 × 0.5–1.3 mm, divergent and patent, dark castaneous with black hues), and from E. marojejyense Tardieu by its darker, black rhizome scales, and by larger, coriaceous (vs. succulent-like), and non-spathulate fronds.

Description:–Rhizomes long-creeping, 1.0–2.5 mm diam., branched, scaly (Figs. 37A–37C); rhizome scales very scattered, appressed, ovate, 0.5–1.2 × 0.2–0.6 mm, sclerotic, dark castaneous to black, opaque, bright, peltate, apices acute to attenuate, margins with some cilia which are contorted, glandular and shorter than the width of the scale body (Fig. 37D).

Sterile fronds erect, inserted in 2 indistinct rows (5–)10–33 mm apart, 10–23 cm long (Fig. 37A); petioles 4–10 cm long, <1.5 mm diam., light brown, with a few scales in the basal half; petiole scales larger than those of rhizomes, up to 1.5 mm long, non-subulate, lanceate to linear, little or not sclerotic, dark castaneous to black, margins with some glandular cilia; sterile laminae coriaceous, lanceolate to narrowly elliptic, 6–13 × 1.5–2.5 cm, bases acuminate long-decurrent in a narrow wing often to the petiole base, apices acute to round, margins clear, cartilaginous and not thinner, both surfaces with inconspicuous scales which are deciduous on adaxial surfaces (Figs. 37E and 37F); laminar scales scattered, non-subulate, irregularly arachnidoid, appressed, 0.3–0.5 mm diam, castaneous (Fig. 37G); median veins prominulous to flat on the abaxial surface (Fig. 37E), prominulous and slightly sulcate with rounded ribs on the adaxial surfaces (Fig. 37F), ending before or at the laminar margin, with sometimes at the base some scales similar to petiole scales; lateral veins often indistinct, free, simple or 1–2-bifurcate.

Fertile fronds about as long as the sterile ones, long-petiolate (½–⅔ of total frond length), laminae narrower, <0.5 cm broad.

Eponymy:–Elaphoglossum sabineanum is named in memory of our good friend and colleague Sabine Comtet-Andriamanjatoarivo (1961–2013), a Malagasy botanist who was technician of the Herbarium P in the Muséum national d’Histoire naturelle (Paris, France); her botanical skills and human qualities were great, rare, and appreciated by everyone.

Habitat and distribution:–Endemic to Madagascar, E. sabineanum is rare and grows as epiphytic in wet evergreen forests in North and Center, in the Eastern and Central phytogeographic domains, 720–1,225 m (Fig. 38).

Additional specimens examined (paratypes):–MADAGASCAR. Perrier de la Bâthie 6130 (P). - Rabarimanarivo 197 (P). - Rakotondrainibe 2571 (P), 2655bis (P), 2655 (P), 4119 (P), 6187 (P), 6374 (P).

Elaphoglossum sinensiumbrarum Rouhan, sp. nov. (Figs. 39 and 40)

Figure 39 Elaphoglossum sinensiumbrarum Rouhan.

(A) Habit. (B) Rhizome. (C) Rhizome scale. (D) Detail of sterile lamina, abaxial surface. (E) Detail of sterile lamina, adaxial surface. (F) Laminar scales. (G and H) Habit in natura. A–H, Rouhan 1385, P02432989. Photos: G. Rouhan/MNHN.

Figure 40 Distribution map of Elaphoglossum sinensiumbrarum.

Red dots represent localities of specimens, and open circles represent the six main cities in Madagascar; the five altitudinal ranges corresponding globally to those generally recognized in Madagascar (Humbert, 1955; Faramalala, 1995) are represented in green (0–400 m), yellow (400–800 m), light brown (800–1,200 m), medium brown (1,200–1,800 m), and dark brown (>1,800 m).

Type:–MADAGASCAR. Alaotra Mangoro, Moramanga, Parc national Mantadia, aux alentours de la piste au départ de PK9, circuit Rianasoa-Chutes sacrées, 18°49′50″S 48°26′07″E, 940 m, 10 Nov. 2011, G. Rouhan, M. Gaudeul, J. Ranaivo 1385 (holotype: P [P02432989!]).

Diagnosis:–Elaphoglossum sinensiumbrarum Rouhan is distinguished from E. lepervanchei (Bory ex Fée) T.Moore by particularly prominulous lateral veins, tangible and visible on both sides, which look like Chinese shadows. This latter character is shared with E. prominentinervulum Rouhan, but E. sinensiumbrarum is quite different with narrowly lanceolate to linear, and clear rhizome scales (vs. ovate to lanceolate, and dark castaneous with sometimes darker spots).

Description:–Rhizomes short-creeping, 2.5–3.5 mm diam., unbranched, densely scaly (Fig. 39B); rhizome scales narrowly lanceolate to linear, 2.5–6.0 × 0.3–1.8 mm, irregularly curved, papyraceous, scarious, concolorous, light brown to stramineous, matte, translucent, bases round, peltate, apices attenuate to angustate, margins with some pluricellular appendages which are ciliform, glandular, contorted and longer than the width of the scale body (Fig. 39C).

Sterile fronds erect, inserted in two distinct rows, up to 8 mm apart, 12–29 cm long (Fig. 39A); petioles 4–13 cm long, 0.6–1.2 mm diam., glabrous with a few scales at base; petiole scales non-subulate, similar to those of rhizomes but smaller and dark brown to black; sterile laminae chartaceous narrowly elliptic, 8–16 × 1.0–2.5 cm, bases and apices cuneate to attenuate, both surfaces with inconspicuous scales which are deciduous on adaxial surfaces (Figs. 39D and 39E); laminar scales scattered, non-subulate, <0.5 mm diam., irregularly arachnidoid, appressed, dark castaneous (Fig. 39F); median veins prominulous on both surfaces, round on the abaxial surface (Fig. 39D), sulcate with rounded ribs on the adaxial surfaces (Fig. 39E); lateral veins simple or 1–2-bifurcate, free, particularly prominulous, tangible and visible on both sides (Figs. 39D, 39E and 39G).

Fertile fronds about as long as the sterile ones, longer-petiolate (ca. 3/10 of total frond length), laminae narrower, <1.3 cm broad.

Etymology:–The specific epithet derives from the Latin adjective, sinensis, Chinese, and name umbrae, shadows; it refers to the lateral veins wich are particularly prominulous and look like Chinese shadows in the wild.

Habitat and distribution:–Endemic to Madagascar, Elaphoglossum sinensiumbrarum is rare and grows as terrestrial or epiphytic in wet evergreen forests in Center, in the Central phytogeographic domain, 940–1,200 m (Fig. 40).

Although being known by only 3 localities far from each other, E. sinensiumbrarum might be more widely distributed because the species could have been mostly ignored in being confused with E. lepervanchei (Bory ex Fée) T.Moore which is one of the most frequent and very variable species.

Additional specimens examined (paratypes):–MADAGASCAR. Barnett 321 (P). - Lastelle s.n. (P01359943). - Rouhan 1410 (P).

Elaphoglossum subglabricaule Rouhan, sp. nov. (Figs. 41 and 42)

Figure 41 Elaphoglossum subglabricaule Rouhan.

(A) Habit. (B and C) Rhizome. (D and E) Rhizome scales. (F) Glutinous dots, abaxial surface. (G) Detail of sterile lamina, abaxial surface. (H) Glutinous dots, adaxial surface. (I) Detail of sterile lamina, adaxial surface. A, C, E, G, I, Rakotondrainibe 2156, P00006369. B, F, H, Rakotondrainibe 2074, P00006299. D, Rabarimanarivo 144, P01588602. Photos: G. Rouhan/MNHN.

Figure 42 Distribution map of Elaphoglossum subglabricaule.

Red dots represent localities of specimens, and open circles represent the six main cities in Madagascar; the five altitudinal ranges corresponding globally to those generally recognized in Madagascar (Humbert, 1955; Faramalala, 1995) are represented in green (0–400 m), yellow (400–800 m), light brown (800–1,200 m), medium brown (1,200–1,800 m), and dark brown (>1,800 m).

Type:–MADAGASCAR. Antsiranana, Andapa, Befingotra, RS d’Anjanaharibe-Sud, sur le versant Sud-Est, à 6.5 km au S-SW du village de Befingotra, 14°45′18″S 49°30′18″E, 870 m, 19 Oct. 1994, F. Rakotondrainibe & F. Raharimalala 2074 (holotype: P [P00006299!]; isotype: TEF).

Diagnosis:–Elaphoglossum subglabricaule, differs from other species with glutinous dots on fronds by short-creeping rhizomes with a few rhizome scales, and closely-spaced fronds (vs. E. humbertii C.Chr.), by fronds without scales (vs. E. patriceanum Rouhan and E. lancifolium (Desv.) C.V.Morton), by herbaceous laminae (vs. coriaceous in E. patriceanum) which are oblanceolate with obtuse to round apices, and smaller and rare to scattered rhizome scales (vs. dense rhizome scales in E. gladiifolium Rouhan and E. patriceanum). Elaphoglossum subglabricaule is clearly distinct from E. glabricaule Rouhan and E. perrierianum C.Chr. (all three species have glabrous or subglabrous rhizomes) by the narrowly oblanceolate laminae (vs. narrowly linear, and linear in E. glabricaule and E. perrierianum, respectively), and by fronds without scales (vs. E. perrierianum).

Description:–Rhizomes short-creeping, 2.5–4.5 mm diam., unbranched, black, glossy, glutinous, subglabrous with rare scales especially at apices (Figs. 41B and 41C); rhizome scales ovate, 0.6–1.0 × 0.2–0.4 mm, sclerotic, dark ferrugineous to black, bright, opaque, bases round, apices acute to obtuse, margins subentire with a few short teeth or cilia (Figs. 41D and 41E).

Sterile fronds erect, inserted in distinct rows, <5 mm apart, 13–54 cm long (Fig. 41A); petioles 2–17 cm long, 0.8–1.6 mm diam., with numerous glutinous dots, ferrugineous to black, and glabrous otherwise or with rare scales similar to those of rhizomes but smaller; sterile laminae herbaceous, narrowly oblanceolate, 11–38 × 1.0–3.5 cm, bases attenuate, apices round to obtuse, both surfaces with numerous glutinous dots ferrugineous, at first bright and translucent, then turning blackish, matte and opaque (Figs. 41F–41I); median veins ending at 3–5 mm from the laminar apices, prominulous on both surfaces, round on abaxial surfaces (Fig. 41G), slightly sulcate with angular ribs on adaxial surfaces (Fig. 41I); lateral veins visible, simple or bifurcate, free, apices enlarged without hydathodes or intramarginal vein.

Fertile fronds unknown.

Etymology:–The specific epithet derives from the Latin prefix sub- (almost), adjective glaber, glabrous, and name caulis, stem; it refers to the almost glabrous rhizomes.

Habitat and distribution:–Endemic to Madagascar, E. subglabricaule is rare and grows as epiphytic in evergreen forests from the North and Center, in Eastern and Central phytogeographic domains, 660–870 m (Fig. 42).

Additional specimens examined (paratypes):–MADAGASCAR. Rabarimanarivo 144 (P). - Rakotondrainibe 2156 (P).

Elaphoglossum tsaratananense Rouhan, sp. nov. (Figs. 43 and 44)

Figure 43 Elaphoglossum tsaratananense Rouhan.

(A) Habit. (B) Rhizome. (C) Rhizome apex with scales. (D) Rhizome scales. (E) Petiole. (F) Petiole scale. (G) Detail of sterile lamina, abaxial surface. (H) Detail of sterile lamina, adaxial surface. (I) Laminar scales. A and B, D–I, Janssen 2912, P00915863. C, Rasolohery 404, P00338390. Photos: G. Rouhan/MNHN.

Figure 44 Distribution map of Elaphoglossum tsaratananense.

Red dots represent localities of specimens, and open circles represent the six main cities in Madagascar; the five altitudinal ranges corresponding globally to those generally recognized in Madagascar (Humbert, 1955; Faramalala, 1995) are represented in green (0–400 m), yellow (400–800 m), light brown (800–1,200 m), medium brown (1,200-1,800 m), and dark brown (>1,800 m).

Type:–MADAGASCAR. Massif de Tsaratanana, montagnes au nord de Mangindrano, Diego, Ambanja, Marotolana, Ampanopia, Ampitsinjovana, 14°8′31″S 48°58′04″E, 2,063–2,300 m, 25 avr. 2001, A. Rasolohery 404 (holotype: P [P00338390!]; isotype: MO [MO-2995834!]).

Diagnosis:–Elaphoglossum tsaratananense is easily distinguished from E. rufidulum (Willd. ex Kuhn) C.Chr., E. multisquamosum Bonap., E. poolii (Baker) Christ, and E. leucolepis (Baker) Krajina ex Tardieu, by its rhizome scales, which are dark castaneous to black, bright (light brown and matte in E. multisquamosum and E. poolii), with glandular marginal cilia (vs. acicular cilia in E. rufidulum and E. leucolepis); furthermore, petiole scales of E. tsaratananense are concolorous (bicolorous in E. rufidulum and E. leucolepis).

Description:–Rhizomes short-creeping and ascending, 3.2–5.0 mm diam., glutinous, all over densely scaly (Figs. 43B and 43C); rhizome scales loosely appressed to patent, lanceate to lanceolate, 2.0–3.5 × 0.5–1.5 mm, dark castaneous to black, bright and brittle, bases cordate to truncate, apices cuneate to acuminate, margins subentire with sometimes some acicular cilia but especially sessil glands and glandular cilia <0.5 mm long, cilia contortate, reddish brown, more abundant towards apices (Fig. 43D).

Sterile fronds erect to spreading, inserted in two barely distinct rows, <5 mm apart, 10–30 cm long (Fig. 43A); petioles 3–13 cm long, 0.8–1.0 mm diam., densely scaly (Fig. 43E); petiole scales patent, non-subulate, lanceate to lanceolate, 2.5–5.0 × 0.7–1.5 mm, light red, matte, translucent, scarious, bases cordate to truncate, apices cuneate, margins ciliate with acicular cilia, 15–30 on each side, thin, 0.2–0.5 mm long (Fig. 43F); sterile laminae chartaceous, narrowly elliptic, 5–16 × 1.0–2.5 cm, bases acute to cuneate, apices acute quite abruptly decrescent, both surfaces and margins densely scaly without completely masking the laminae, adaxial surface glabrescent (Figs. 43G and 43H); laminar scales scattered to imbricate, non-subulate, lanceate to lanceolate, 2.0–4.0 × 0.5–1.0 mm, light red, translucent, scarious, bases hemi-infundibular, apices acute to cuneate, margins ciliate with acicular cilia, 10–25 cilia on each side, 0.4–1.0 mm long and sometimes longer than the width of scale bodies (Fig. 43I); median veins prominulous, round on the abaxial surface, flat and slightly sulcate on the adaxial surfaces; lateral veins slightly prominulous on the abaxial surface, simple or 1–2-bifurcate, free.

Fertile fronds as long as or slightly longer than the sterile ones; petioles ½– ⅗ of total frond length; fertile laminae narrower, 1.0–1.3 cm broad, bases truncate-asymmetric.

Etymology:–The specific epithet derives from the Malagasy name “Tsaratanana”, highest massif of Madagascar, integral nature reserve, and single locality known for the species; Tsaratanana is derived from “tsara”, beautiful, good, generous, and “tanàna”, town, village, hamlet, and the whole word means the beautiful village (Lalao Andriamahefarivo, pers. com., 2020).

Habitat and distribution:–Endemic to Madagascar, E. tsaratananense is rare and grows as terrestrial (or epiphytic at the base of the trees) in mountain sclerophyll forests on ridges and summits, in the North, in the Central phytogeographic domain, 2,063–2,332 m (Fig. 44).

Additional specimens examined (paratypes):–MADAGASCAR. Janssen 2899 (P), 2912 (P). - Razafitsalama 272 (P).

Elaphoglossum viridicaule Rouhan, sp. nov. (Figs. 45 and 46)

Figure 45 Elaphoglossum viridicaule Rouhan.

(A) Habit. (B) Rhizome. (C) Rhizome (arrows indicate the aerophores). (D) Rhizome scales. (E) Detail of sterile lamina, abaxial surface. (F) Laminar scales. (G and H) Habit in natura. A, Rakotondrainibe 4744, P00134812. B, Rouhan 319, P00749262. C, Rouhan 317, P00749259. D–F, Rouhan 407, P00749370. G, Rouhan 374, P00749331. H, Antilahimena 7619, P06488839. Photos: G. Rouhan/MNHN.

Figure 46 Distribution map of Elaphoglossum viridicaule.

Red dots represent localities of specimens, and open circles represent the six main cities in Madagascar; the five altitudinal ranges corresponding globally to those generally recognized in Madagascar (Humbert, 1955; Faramalala, 1995) are represented in green (0–400 m), yellow (400–800 m), light brown (800–1,200 m), medium brown (1,200–1,800 m), and dark brown (>1,800 m).

Type:–MADAGASCAR. Région de Alaotra-Mangoro, district de Moramanga, Andasibe, Réserve d’Analamazaotra, 18°56′16″S 48°25′14″E, 950 m, 11 Nov. 2004, G. Rouhan & T. Janssen 407 (holotype: P [P00749370!]; isotypes: NBG, NY).

Diagnosis:–Elaphoglossum viridicaule differs from the three other species with green rhizome apices in natura, black rhizome scales, and subglabrous fronds (E. rakotondrainibeae Rouhan, E. longiacuminatum Rouhan et E. approximatum Rouhan), by its rhizome scales which are sclerotic, shorter, and most often with a few marginal glandular cilia; E. viridicaule differs furthermore from E. rakotondrainibeae and E. longiacuminatum by short-creeping rhizomes with closely spaced fronds.

Description:–Rhizomes short-creeping, 3.2–6.0 mm diam., branched (branches short and often numerous, with many short to non-developping buds), showing green apices in natura, with scattered scales (Fig. 45B); rhizome scales black, opaque, bright, sclerotic and brittle, peltate, bases round, apices attenuate to angustate, margins with brown and matte glandular cilia; smallest scales always present, appressed, ovate to lanceolate, 0.3–1.0 × 0.2–0.7 mm, but largest scales brittle and deciduous, patent, narrowly lanceate to linear, to 3.0 × 0.4 mm long (Fig. 45D); aerophores present in pairs at the frond base, finger-shaped, green (Fig. 45C).

Sterile fronds erect, inserted in 2 or 3 indistinct rows, <5 mm apart, 10–50 cm long (Fig. 45A); petioles 2–15 cm long, 1.0–2.7 mm diam., glabrous or with, at base, a few deciduous scales similar to the largest rhizome scales; sterile laminae coriaceous, elliptic, 8–35 × (1.3–)2.5–6.0 cm, bases and apices cuneate to short-acuminate, both surfaces with inconspicuous scales (Fig. 45E); laminar scales deciduous, scattered, non-subulate, <1 mm diam., irregularly arachnidoid, appressed, dark castaneous to black; median veins prominulous on both surfaces, round on abaxial surfaces, sulcate with rounded ribs on adaxial surfaces; lateral veins barely or not visible, simple or 1–2-bifurcate, free, vein apices sometimes anastomosed into a discontinuous intramarginal vein.

Fertile fronds about as long as the sterile ones, petioles as long as or longer than the sterile ones (½–⅗ of total frond length), laminae slightly narrower; sporangia not reaching margins, leaving all around a narrow, marginal strip.

Etymology:–The specific epithet derives from the Latin adjective viridis, green, and name caulis, stem; it refers to the green rhizomes apices.

Note:–Elaphoglossum viridicaule resembles the Mauritian endemic E. ×revaughanii Lorence, a hybrid possibly between E. lepervanchei (Bory ex Fée) T.Moore and E. sieberi (Hook. & Grev.) T.Moore. However, E. viridicaule is supported as a distinct species, as firstly, one of the putative parent (E. sieberi) is not known in Madagascar, and secondly, the hybrid E. ×revaughanii, is known only by 2 small vegetatively reproducing clonal populations (Lorence, 1984; Lorence & Rouhan, 2004; Lorence & Rouhan, 2008) thus excluding a dispersion by spores of this hybrid from Mauritius to Madagascar.

Habitat and distribution:–Elaphoglossum viridicaule is frequent in Madagascar and occurs in Moheli in the Comoros; it grows as epiphytic or epilithic in evergreen forests from North to South in Madagascar, in the Eastern and Central phytogeographic domains, 10–1,560 m (Fig. 46).

Additional specimens examined (paratypes):–MADAGASCAR. Antilahimena 3269 (P), 5392 (P), 7619 (P). - Boivin 1581/44 (P). - Corréard s.n. (P00611006). - Cours 204 (P), 4037 (P). - Decary 16777 (P), 17678 (P), 17689 (P), 17816 (P), 18088 (P), 18097 (P). - Janssen 2840 (P). - Perrier de la Bâthie 6128 (P), 6141 (P). - Proisy 239 (P). - Rabarimanarivo 104 (P). - Rakotondrainibe 727 (P), 1958 (P), 1959 (P), 1960bis (P), 1960 (P), 1963 (P), 1965 (P), 1968 (P), 2080 (P), 2239 (P), 2691 (P), 2962 (P), 2991 (P), 3025 (P), 3411 (P), 3725 (P), 3838 (P), 4744 (P), 5079 (P), 5804 (P), 5885 (P), 5951 (P), 6291 (P). - Rakotovao 1537 (P). - Randriatsivery 556 (P). - Rasolohery 224 (P), 318 (P), 485 (P), 631 (P), 782 (P). - Ratovoson 439 (P). - Razakamalala 1150 (P), 4194 (P). - Rouhan 329 (P), 374 (P), 390 (P), 433 (P), 479 (P), 485 (P).

This article is dedicated to the memory of our good friend and colleague Professor Jean-Noël Labat (1959–2011), former Head of the national Paris Herbarium (P). Although being an expert of the Malagasy Angiosperm Flora himself, Jean-Noël, as an advisor, had strongly supported my PhD on the systematics of Elaphoglossum, and became a most inspiring person for my career as a botanist. I express my warm thanks to my first Elaphoglossum mentors, Robbin Moran and John Mickel. Constructive peer reviews of Paulo Labiak, an anonymous reviewer, and the editor Victoria Sosa, made significant contributions to the manuscript. Collecting permits in Madagascar were granted by Madagascar National Parks and the Ministère de l’Environnement, de l’Ecologie et des Forêts. We are grateful, for field assistance, to CNRE-Madagascar and MBG-Madagascar, to Emile Randrianjohany and Jaona Ranaivo (CNRE), Dr. Hery Lisy Ranarijaona (Mahajanga University), Dr. Lucie Bauret, Dr. Myriam Gaudeul, Dr. Thomas Janßen, Dr. France Rakotondrainibe (MNHN, by the time), Patrice Antilahimena, Charles Rakotovao, and Désiré Ravelonarivo (MBG), Tahina Razafindrahaja and Hanta Razafindraibe (PBZT), and last but not least to Dr. Sylvain Razafimandimbison (Stockholm Museum of Natural History). We are grateful to the curators and staffs of herbaria B, BM, BR, G, K, MO, NBG, NY, P, PR, PRE, TAN, TEF, US, for their work and assistance with the collections. I am thankful to Dr. Catherine Reeb for the distribution maps and to Lalao Andriamahefarivo (MBG) and Pierrot Rabenandrasana for tracing the origins of some Malagasy words I am thankful to Dr. Myriam Gaudeul and Fabrice Clermont for sharing an office full of fern specimens, and to Myriam for her openness to have started to collaborate on Malagasy ferns.

Additional Information and Declarations

Competing Interests

Author Contributions

Field Study Permissions

Data Availability

New Species Registration

The author declares that he has no competing interests.

Germinal Rouhan conceived and designed the experiments, performed the experiments, analyzed the data, prepared figures and/or tables, authored or reviewed drafts of the paper, and approved the final draft.

The following information was supplied relating to field study approvals (i.e., approving body and any reference numbers):

Collecting permits in Madagascar were granted by Madagascar National Parks and the Ministère de l’Environnement, et du Développement Durable (project numbers: 70/19/MEDD/SG/DGF/DSAP/SCB.Re, and 207/15/MEEMF/SG/DGF/DAPT/SCBT, and 199/15/MEEMF/SG/DGF/DAPT/SCBT, and 241/11/MEF/SG/DGF/DCB.SAP/SCB).

The following information was supplied regarding data availability:

Data is available at MNHN-Science: P00006299, P00006305, P00006354, P00006369, P00006427, P00006460, P00006476, P00046645, P00046832, P00046836, P00046838, P00046839, P00046850, P00046857, P00046862, P00046890, P00046910, P00046927, P00046938, P00046946, P00046948, P00046961, P00059743, P00059750, P00059778, P00059834, P00059835, P00059888, P00059937, P0006293, P00067015, P00067053, P00067056, P00067084, P00067085, P00067086, P00067121, P00067229, P00067230, P00084552, P00084578, P00084692, P00084711, P00084887, P00084890, P00084950, P00084966, P00084983, P00084986, P00084987, P00084993, P00085061, P00085069, P00085089, P00085137, P00085138, P00085152, P00085153, P00085177, P00085186, P00085187, P00085215, P00085223, P00085229, P00085230, P00085271, P00085526, P00085744, P00085816, P00085817, P00134089, P00134130, P00134348, P00134812, P00134817, P00134952, P00151009, P00151010, P00151011, P00151018, P00179407, P00179566, P00179605, P00179606, P00179627, P00179654, P00181029, P00181037, P00181083, P00181101, P00181107, P00181114, P00181115, P00181119, P00181156, P00181178, P00181187, P00181192, P00181216, P00181220, P00212294, P00212366, P00212368, P00212379, P00212429, P00212461, P00212462, P00212538, P00212659, P00212739, P00212751, P00212754, P00233415, P00233417, P00233419, P00244836, P00244881, P00244936, P00244937, P00248502, P00248587, P00248588, P00248594, P00248595, P00248637, P00248640, P00248650, P00262130, P00327948, P00327983, P00328049, P00338390, P00338392, P00338393, P00338407, P00338410, P00338411, P00411565, P00411566, P00411568, P00411619, P00590896, P00590915, P00590927, P00590949, P00590953, P00605311, P00605313, P00605314, P00605315, P00605317, P00605318, P00605319, P00605320, P00605321, P00610922, P00611002, P00611006, P00611017, P00611018, P00611019, P00611058, P00611099, P00611103, P00611104, P00611189, P00611190, P00611337, P00611338, P00611339, P00611340, P00611341, P00612807, P006135, P006137, P006143, P006144, P006149, P006150, P006151, P006152, P006154, P006155, P006157, P006158, P006159, P006160, P00636590, P00636591, P00636594, P00636600, P00636601, P00636602, P00749222, P00749227, P00749228, P00749239, P00749240, P00749241, P00749242, P00749243, P00749245, P00749250, P00749263, P00749264, P00749266, P00749267, P00749268, P00749269, P00749276, P00749298, P00749306, P00749307, P00749308, P00749309, P00749312, P00749316, P00749331, P00749333, P00749335, P00749336, P00749348, P00749370, P00749373, P00749374, P00749380, P00749400, P00749455, P00749463, P00749464, P00749465, P00749466, P00915541, P00915747, P00915748, P00915749, P00915750, P00915751, P00915752, P00915759, P00915863, P01359930, P01359932, P01359933, P01359936, P01359937, P01359943, P01359952, P01359956, P01359957, P01359968, P01359970, P01359971, P01359974, P01414143, P01556618, P01566071, P01566072, P01588596, P01588597, P01588598, P01588599, P01588600, P01588602, P01588603, P01588604, P01588605, P01588606, P01588607, P01588608, P01588609, P01588610, P01588611, P01588613, P01588619, P01616822, P01616842, P02432650, P02432668, P02432690, P02432715, P02432716, P02432749, P02432750, P02432772, P02432830, P02432831, P02432856, P02432989, P02433028, P02434953, P02434961, P02435014, P02435015, P02435062, P02435070, P02435133, P02435163, P02435813, P02435814, P02435817, P02435818, P02435917, P06140157, P06142046, P06488350, P06488351, P06488358, P06488362, P06488532, P06488838, P06488839, P06488852, P06489119, P06489121, P06489127, P06489128, P06489129, P06489130, P06489131, P06489132, P06489133, P06489135, P06489136, P06489139, P06489140, P06489141, P06489142, P06489145.

The following information was supplied regarding the registration of a newly described species:

Elaphoglossum ambrense Rouhan: 77212897-1

Elaphoglossum andohahelense Rouhan: 77212898-1

Elaphoglossum anjanaharibense Rouhan: 77212955-1

Elaphoglossum approximatum Rouhan: 77212956-1

Elaphoglossum brachymischum Rouhan: 77212957-1

Elaphoglossum cerussatum Tardieu subsp. brunneum Rouhan: 77212958-1

Elaphoglossum coracinolepis Rouhan: 77212959-1

Elaphoglossum desireanum Rouhan: 77212960-1

Elaphoglossum glabricaule Rouhan: 77212961-1

Elaphoglossum gladiifolium Rouhan: 77212962-1

Elaphoglossum leucolepis (Baker) Krajina ex Tardieu subsp. nanolepis Rouhan: 77212963-1

Elaphoglossum leucolepis (Baker) Krajina ex Tardieu subsp. nigricans Rouhan: 77212964-1

Elaphoglossum longiacuminatum Rouhan: 77212965-1

Elaphoglossum patriceanum Rouhan: 77212966-1

Elaphoglossum perangustum Rouhan: 77212967-1

Elaphoglossum prominentinervulum Rouhan: 77212968-1

Elaphoglossum rakotondrainibeae Rouhan: 77212969-1

Elaphoglossum repandum Rouhan: 77212970-1

Elaphoglossum sabineanum Rouhan: 77212971-1

Elaphoglossum sinensiumbrarum Rouhan: 77212972-1

Elaphoglossum subglabricaule Rouhan: 77212973-1

Elaphoglossum tsaratananense Rouhan: 77212974-1

Elaphoglossum viridicaule Rouhan: 77212975-1.

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
