# Peer review of "Synoptic revision of the fern genus Elaphoglossum Schott ex J.Sm. (Dryopteridaceae) in Madagascar, with the description of 23 new taxa, all but one endemic"

_PeerJ, doi:10.7717/peerj.10484_

## Round 0.1 · original submission · Minor Revisions

Both reviewers coincided as well as myself that your paper needs minor changes, some of them in Abstract, to make clear the contribution of your paper and presentation of distribution maps as well as some changes in the identification keys. There are also very detailed reviews in the two files that the reviewers included.

·

Basic reporting

no comment

Experimental design

no comment

Validity of the findings

no comment

Additional comments

In the following, I present my review on the manuscript "Synoptic revision of the fern genus Elaphoglossum Schott ex J.Sm. (Dryopteridaceae) in Madagascar, with the description of 23 new taxa, all but one endemic", authored by Dr. Germinal Rouhan.

The manuscript presents a treatment for 55 species from Madagascar, almost half of the newly described in this work. It is an excellent contribution to the Flora of Madagascar and adds valuable information to the genus' diversity as well. The manuscript is well written, and the new species are described following the rules of the International Code of Botanical Nomenclature. The author did a great job illustrating the morphology of new taxa, as well as their distribution in Madagascar. I have made all my suggestions in the manuscript, marked as track changes. Some aspects that I judge more critical I highlight below:

Abstract
1- Some changes in the abstract could make it easier for the readers to capture the main points of your manuscript (see my suggestions in the marked pdf.).
2- Still in the abstract, it is said that "six novel dichotomous keys are proposed...". I found only four keys – two of them were combined into a single one. Please, revise.
Introduction
3- The reference for the Leipzig Catalogue of Vascular Plants is missing.
4- Lines 86-87: Please, check the correct citation for those references.
5- Line 92: I suggest changing the sentence "The taxonomy of Elaphoglossum is challenging because the genus is distinctive but, at first sight..." by something like "Although the genus is readily distinguished among the ferns, most of its species are morphologically uniform and difficult to distinguish."
6- Lines 101-104. I think you could also cite here Vasco et al. (2009) on section Lepidoglossa, and Matos et al. (2019) on Section Polytrichia.
Material and Methods
7- In the distribution maps, there are dots with distinct colors, some are solids, and some are not. Could you explain what they are? I also think this explanation should be included in the captions, for the ease of understanding.
Keys
8- I suggest using a "noun-adjective" format, as done for other couplets in the key. For example, in couplet 1, I would replace the sentence "... inconspicuously reduced scales (< 0.5 mm in diam.), arachnoid, scattered, dark brown to black..." by "but with scales that are inconspicuous, reduced (...), arachnoid, ... "and so on.
9- Couplet 2a: I suggest replacing "Frond scales" with "Laminar scales." This is how it is described for all species.
10- As for the subulate scales, I suggest describing them as "hairlike or conical, with its margins inrolled for most of their length."
11- Couplet 2b: I noticed that in some descriptions, like in Elaphoglossum ambrense, there is no mention of whether the scales are subulate or not. I think it is essential to mention in the description of any character that has been used in the key, even if it is missing in a particular species.
12- I do not understand why the sections I and II were combined into a single key. Is there a particular reason for that? If so, I suggest it be explained what the reason to combine them (as you did for Sections III and IV, lines 270-271) is.
Descriptions and comments
13- I think the figures need to be cited in more detail. Could you cite the particular figures (e.g., Fig. 1 A) in the diagnosis you provided, or in the descriptions? This would render the figure citation more precise and accessible to the readers.
14- As for the Holotypes in Paris, I suggest citing them as "holotype: P [P00749250]". I think it is important to keep the Herbarium acronym just as "P", and mention the barcode number between brackets.
15- In the descriptions, I would start mentioning the habit for each species.
16- Please, cite the maps right after your comment on "Habitat and Distribution."
17- In several instances, there are Isotypes in other Herbaria. Could you provide the barcodes for those specimens as well?
18- When you provide the diagnosis for your new taxa, sometimes you compare them with other similar species. In this case, it is important to mention not only the characters of the new species but also of the species that you are comparing it to. For instance, in the diagnosis for Elaphoglossum brachymischum you compare it to E. pseudovillosum and say that "Elaphoglossum brachymischum clearly differs by short-petiolate fronds, rhizome scales clearer, shorter and narrower, laminae thinner and herbaceous, and laminar scales with several long, ciliform, marginal appendages above the base." Please, include here how these characters are in E. pseudovillosum as well.
19- Elaphoglossum sabineanum: It is just a suggestion, but why not call it E. sabineae?

Figures:
20- Please, check whether all the letters in the figure captions are in parenthesis.
21- Remember to include in the captions an explanation of why the dots are different on the same map.

Reviewer 2 ·

Basic reporting

The manuscript “Synoptic revision of the fern genus Elaphoglossum Schott ex J.Sm. (Dryopteridaceae) in Madagascar, with the description of 23 new taxa, all but one endemic” is a much needed addition to the taxonomy of one of the most complicated fern genus. Elaphoglossum, with more than 600 species, is one of the most complex taxonomically and less studied fern genera. The revision is built on accumulated knowledge of an authority in the group, in a very diverse and understudied area.

The figures are excellent, with images illustrating well all the relevant characters for species recognition in the genus. They are high quality images, well displayed and labelled.
The keys are overall very good, although I suggest making separate keys to each section. A few issues listed below:
In the key to the sections (ln 232), It would be better, if possible, to put a couplet to separate Elaphoglossum section Elaphoglossum and E. section Squamipedia;
In the key to section Polytrichia and Setosa (ln 252): to make this step comparable with the opposing one in the couplet, describe in which parts of the plant these scales occur.
Other minor comments about the keys were made in the file
Regarding the descriptions, they are well standardized, very objective, and well-written. The only information that is not consistent throughout species descriptions is about sporangia. I also think it would be relevant to describe and illustrate the spores of the newly described species, the only information I believe is missing in the description. Although not mandatory, for the subspecies, I think a short description in addition to the diagnosis should be provided. Some suggestions were made in a few species diagnoses. For Elaphoglossum repandum, the diagnosis, which is mandatory, is missing.

Experimental design

As a taxonomic paper describing new taxa, the manuscript question is well-defined, with appropriately described methods

Validity of the findings

The description of so many new taxa, most of them endemic to Madagascar, is a major step in helping preserving plant diversity in a biodiversity hotspot. The keys also are essential to help identify and catalogue the diversity.

Additional comments

The manuscript is well-written, and I have made overall small suggestions or corrections, but I would like to point out I am not a native speaker. The literature cited is well referenced and relevant for the manuscript. Other general minor issues are highlighted below:
Ln 76: There are six numbers but only seven listed sources.
Ln 106: I think this should rephrased to be more assertive. Based on the currently known richness, it either is or isn’t the second center or diversity
Ln 107: Maybe state here how many or which fraction of overall described species occur in Madagascar
Ln 157: QGis reference missing here
Lns 159-160: I recommend this information to be inserted in the maps as a legend
Ln 184: It is not clear what the author meant here, I suggest rephrasing
Ln 197: It is interesting to present here also the number of endemic species
Ln 607: Since it’s the diagnosis, even if it gets redundant, it would be interesting to cite those species here
Ln 564: I do not know political geography of Madagascar, so I highlighted here for the author to check if it’s not erroneously repeated
Ln 569: Maybe “makes” it’s a more appropriate verb here
Ln 770: I believe “similar to” could be more appropriate for the diagnosis
Ln 893: I think this is not a very useful information for the diagnosis

Annotated reviews are not available for download in order to protect the identity of reviewers who chose to remain anonymous.

---

## Round 0.2 · accepted · Accept

Thank you for your corrections. My only concern is in the maps, the red dots are not clearly seen, mostly in records that are close to the limits of the island. Maybe you can change the symbol to triangles and you can upload these figures when reviewing the proofs.